



# Global aeolian dust variations and trends: a revisit of dust event and visibility observations from surface weather stations

Xin Xi

Department of Geological and Mining Engineering and Sciences, Michigan Technological University, Houghton, MI

**Correspondence:** X. Xi (xinxi@mtu.edu)

**Abstract.** This study revisits the use of horizontal visibility and manually reported present weather ($ww$) records from the NOAA Integrated Surface Database (ISD) for characterizing the aeolian dust variability and recent trends over the globe and three largest source regions (North Africa, Middle East, and East Asia). Due to its qualitative nature, $ww$ is combined with visibility to derive a new variable, $VI$, which has higher correlations with the dust emission and burden from satellite

observations and global aerosol reanalyses than does the dust event frequency ($FR$) derived from $ww$ only. Both $FR$ and $VI$ capture the intensive dust activity associated with the prolonged North American drought during the 1950s and Sahelian drought during the 1980s. Correlation analysis suggests soil moisture has a lagged effect on the global dustiness, with a maximum $r = -0.3$ when soil moisture leads $VI$ by 14 months. Through a critical assessment of the $ww$ continuity and $ww$-visibility consistency of various report types in ISD, the SYNOP data are used for global dust trend detection from 1986 to

2019. Globally, $FR$ and $VI$ decreased at a rate of $-0.23\%$ yr$^{-1}$ and $-8.0\times10^{-4}$ km$^{-1}$ yr$^{-1}$, respectively, from 1986 to 1996/97 when dust reached a minimum, followed by a slower rebound at a rate of $0.085\%$ yr$^{-1}$ and $1.9\times10^{-4}$ km$^{-1}$ yr$^{-1}$, respectively. The nonlinear behavior of global dustiness is qualitatively consistent with satellite observations and global aerosol reanalyses. Regionally, North Africa experienced increased dust activity during the past decade after staying below average for most of the 1990s–2000s, in response to reduced soil moisture and increased wind speed following the transition of North Atlantic

Oscillation (NAO) from strong negative to recurring positive phases since 2011. In the Middle East, dust has been increasing since 1998 due to a prolonged drought in the Tigris-Euphrates basin associated with strong negative Pacific Decadal Oscillation (PDO) phases. As PDO turned positive and weak negative after 2015, the amelioration of drought has led to decreased dust activity in recent years. The dust variability in East Asia is primarily driven by wind speed, which explains the dust decline from 1986 to 1997, and the absence of dust trends during the past two decades. This study constitutes an initial effort of creating

a homogenized weather station-based dust-climate dataset in support of wind erosion monitoring, dust source mapping, and dust-climate analysis at local to global scales.

## 1 Introduction

Mineral dust is of major interest to multidisciplinary research related to its interactions with the climate, ocean biogeochemistry, air quality, and agriculture. As a by-product of aeolian processes, dust is emitted from dry, exposed land surfaces, which are

mostly located in arid and semi-arid regions, and is dispersed downwind, sometimes across oceans and continents, before



being removed by gravitational settling and precipitation scavenging. Once airborne, dust particles can cause direct and indirect effects on the radiative energy balance (Sokolik et al., 2001). Dust outflow from continental sources is considered a primary source of mineral micro-nutrients to the open ocean, where the biological productivity is limited by the availability of iron (Mahowald et al., 2009). Through light absorption, dust deposited on snow can decrease the surface albedo with consequences
on the surface energy balance and hydrology (Painter et al., 2010). For human settlements located downwind, airborne dust is a major safety and health hazard due to the elevated particulate matter concentration and dust-borne pathogens (Griffin et al., 2001). Once considered as a natural process, dust emission is increasingly linked with human-induced desertification due to unsustainable land management (e.g., over-irrigation, overgrazing, monocropping, deforestation), which reduces the food production potential and co-benefits (e.g., biodiversity, carbon sequestration) of agricultural lands (Mirzabaev et al., 2019). To
quantify the multifaceted dust impact, it requires fundamental knowledge about the dust source locations and properties, dust variability and trends at different timescales, as well as the physical processes and factors governing the dust variability and trends.

The dust cycle is highly sensitive to the climate, which regulates the supply of erodible materials, the erosive force of surface winds, and the transport route and residence time of dust particles in the air. The dust-climate connection exists on multiple
timescales from seasonal to interannual, multidecadal, or even glacial-interglacial scales. Indeed, dust records recovered from loess deposits, marine sediments or ice cores provide one of the most important means to infer the earth's climate history and test the capability of earth system models for simulating the natural climate variability of the geologic past (Kohfeld and Harrison, 2001). For contemporary dust, satellite remote sensing, especially low- and moderate-resolution passive sensors in sun-synchronous orbits, has greatly advanced the monitoring capability of large-scale dust events on the daily basis. The
longest global operational aerosol product is the over-ocean aerosol optical depth (AOD) climate data record generated from the Advanced Very High Resolution Radiometer (AVHRR) sensors onboard NOAA's weather satellite series beginning in the late 1970s (Zhao et al., 2008). For the first time, the absorbing aerosol index (AAI) from Total Ozone Mapping Spectrometer (TOMS), which is a semi-quantitative measure of column aerosol burden, allowed dust detection over the source area and the derivation of a global dust source map which has been widely used in dust models (Ginoux et al., 2001; Prospero et al.,
2002). In the past two decades, new-generation instruments designed specifically for measuring the atmospheric composition with improved sensor characteristics, most notably the twin Moderate Resolution Imaging Spectroradiometer (MODIS) sensors onboard Terra and Aqua, have further expanded the capability of dust source mapping and characterization from space (Ginoux et al., 2012). The UV and visible-based satellite aerosol records are being extended by a number of successor instruments in sun-synchronous and geostationary orbits (Torres et al., 2007; Jackson et al., 2013; Lindfors et al., 2018).

In general, satellite data are limited by the relatively short record length for characterizing interannual to multidecadal dust variations governed by low frequency climate variability, especially over dust source areas where quantitative information of the column aerosol burden is available only for the last two decades. Dust retrieval over source areas using backscattered light remains a challenging task due to the high surface reflectivity in visible wavelengths. Early sensors, like TOMS, were able to detect elevated dust layers over land in the UV channel, where the surface reflectivity is generally low and nearly lambertian
(Herman and Celarier, 1997). More advanced sensors (such as MODIS) allow the use of blue wavelengths, where the land





surface reflectance is low enough, to expand aerosol retrieval to bright desert surfaces, but with higher uncertainty than over ocean and vegetated areas (Hsu et al., 2004, 2013). Other limitations of using passive satellite sensors for long-term dust analysis include the difficulty of separating dust from the total aerosol signal, limited sampling frequency of polar-orbiting satellites (usually once per day), incapability of detecting dust under clouds, and sensor calibration instability.

Prior to the advent of satellite remote sensing, surface synoptic weather observations, specifically the report of dust event occurrence and visibility, were a primary source of information for monitoring wind erosion and dust activity around the world (e.g., Middleton, 1984; Gillette and Hanson, 1989; McTainsh et al., 1989; Goudie and Middleton, 1992; Qian et al., 2002; Kurosaki and Mikami, 2003; O'Loingsigh et al., 2014). Weather station infrastructure and data systems were established worldwide to meet the needs of weather forecast, aviation operation, and climate analysis. The original weather data may be

stored in different formats and repositories, and vary in the observation schedule, observed variables, instrumental method (e.g., manual vs. automatic), as well as the report protocol depending on the purpose of measurements. To expedite the data acquisition and usage by end users, NOAA has undertaken substantial efforts to merge more than 100 different data sources with over 30,000 stations into a single, complete global archive of surface climate data, called Integrated Surface Database (ISD) (Smith et al., 2011, see https://www.ncdc.noaa.gov/isd). ISD consists of sub-daily observations of meteorological parameters

from tens of thousands of stations worldwide, including pressure, temperature, wind, precipitation, present weather, visibility, and others. The sub-daily data are suitable for the study of climate variations, climate extremes, or even individual meteorological events (such as dust). The development of ISD involves a series of automated quality control applied to each raw record that focuses on the data completeness, extreme value validity, temporal continuity (e.g., spikes in continuous variables), and external consistency (e.g., no snow at well above freezing temperature), which were described in Lott (2004). The automatic

quality control focuses on data standardization and completeness, but does not guarantee the data quality required by specific applications. As a result, additional quality control and homogenization efforts are needed to remove unwanted data artifacts and reveal the true statistical features of the variable of interest. For instance, the UK Met Office Hadley Center developed a dataset for investigating climate extremes by selecting a subset of ISD stations with sufficient record length and observation frequency, and conducting comprehensive intra- and inter-station statistical tests in order to retain the true extreme behaviors

of climate variables (Dunn et al., 2012).

    To use surface weather observations for long-term dust analysis, a major challenge is the qualitative nature of present weather reports, and potential data discontinuity resulting from changes of station coverage over time and in some cases, changes of dust weather definitions, observation method, and reporting protocol (O'Loingsigh et al., 2014). Previously, Mahowald et al. (2007) used the visibility data from a few hundred stations located in dust-dominated regions to investigate global dust variations from

1974 to 2003. By extending the analysis to 1973–2012 and the global SYNOP station network, Shao et al. (2013) used the dust event reports and visibility to derive the near-surface dust mass concentrations, and investigated the global and regional dust trends and climate drivers. This study continues the work of Mahowald et al. (2007) and Shao et al. (2013) by using the present weather and visibility data from ISD to examine global and regional dust variations and trends during recent decades. This study includes a first attempt of continuity and consistency checks of present weather and visibility observations for the

purpose of tracking long-term dust changes, and presents an analytic framework which can be expanded and optimized in the



**Table 1.** Description of dust events in the present weather report from manned stations ($ww$). The number of dust events ($N_{du}$) and harmonic mean visibility (km) of each event category are based on the global SYNOP data between 1973 and 2019.

| $ww$ | Dust code definition | $N_{du}$ | Visibility |
|------|----------------------|----------|------------|
| 06 | Widespread dust in suspension in the air, not raised by wind at or near the station at the time of observation | 1,401,947 | 3.7 |
| 07 | Dust or sand raised by wind at or near the station at the time of observation, but not well-developed dust whirl(s) or sand whirl(s), and no dust storm or sandstorm seen | 617,121 | 3.7 |
| 08 | Well-developed dust or sand whirl(s) seen at or near the station during the preceding hour or at the time of observation, but no dust storm or sandstorm | 52,987 | 5.6 |
| 09 | Dust storm or sandstorm within sight at the time of observation, or at the station during the preceding hour | 36,052 | 3.4 |
| 30 | Slight or moderate dust storm or sandstorm – has decreased during the preceding hour | 32,636 | 1.9 |
| 31 | Slight or moderate dust storm or sandstorm – no appreciable change during the preceding hour | 60,783 | 1.6 |
| 32 | Slight or moderate dust storm or sandstorm – has begun or has increased during the preceding hour | 41,432 | 1.2 |
| 33 | Severe dust storm or sandstorm – has decreased during the preceding hour | 11,630 | 1.0 |
| 34 | Severe dust storm or sandstorm – no appreciable change during the preceding hour | 8,852 | 0.4 |
| 35 | Severe dust storm or sandstorm – has begun or has increased during the preceding hour | 14,084 | 0.6 |
| 98 | Thunderstorm combined with dust/sandstorm at time of observation | 6,089 | 1.5 |

future to create a homogenized global weather station-based dust climatology. This manuscript is organized as follows. Section 2 describes the present weather and visibility variables, and how they are used to quantify the dust burden. Section 3 discusses the continuity and consistency checks to guide the selection and interpretation of weather station data. Section 4 presents a global analysis of decadal mean and interannual dust variations based on the ISD dust records, aerosol reanalysis, and satellite data, followed by more thorough analysis for three most important source regions (North Africa, Middle East, and East Asia) in Section 5, including North Africa, Middle East, and East Asia. The summary of this study are given in Section 6.

## 2 Data

### 2.1 Present weather

Present weather includes qualitative reports of phenomena observed in the atmosphere based on instrumental and/or manual methods, such as precipitation, visibility obstruction, squalls, tornado, and dust storms (WMO, 2018). Present weather reports from manned stations (abbreviated as $ww$ by the World Meteorological Organization, WMO) follow the WMO code table 4677, wherein $ww$ varies from 00 (lowest priority) to 99 (highest priority), each describing the visual perception of weather phenomenon occurring in the reporting period (usually the preceding hour of the time of observation). The order of priority





determines which event to record in $ww$ if multiple events are observed. According to the WMO table 4677, dust events are ranked after fog (40–49) and precipitation (50–99) weather groups, and described by 11 numeric codes: $ww$ = 06–09, 30–35, 98, which are given in Table 1. The dust codes describe the near surface dust conditions with varying intensity (i.e., slight, moderate, or heavy), proximity to the station (i.e., at the station, in the vicinity of the station, or distant from the station), and development stage (i.e., decrease, no appreciable change, or increase). Specifically, $ww$ = 06 describes dust in suspension phenomena at the station caused by dust transport from non-local sources. $ww$ = 07 describes blowing dust raised at or near the station. $ww$ = 08 describes dust devils or swirls which are typically short-lived, local events over hot, dry and flat surfaces. $ww$ = 30–35 describe slight/moderate and severe dust storms. Based on global SYNOP data between 1973 and 2019, dust in suspension ($ww$ = 06) is the most frequently reported (61%), followed by blowing dust ($ww$ = 07, 27%), while severe dust storms ($ww$ = 33–35) account for only 1.5%. Table 1 also shows the harmonic mean visibility associated with each dust code, as a quantitative measure of the ambient dust burden.

In ISD, there are up to 7 $ww$ reports at any time of observation. Here, only the code of the highest priority is retained by following the $ww$ reporting protocol, except that if a dust event is reported, the dust code is retained instead, such that dust events are not superseded by other higher-priority weather groups. To convert the categorical $ww$ reports into quantitative measures of dust activity, earlier studies calculated the dust day frequency as the number of dusty days in a year, wherein a dusty day is defined as the day during which at least 1 dust code is reported (Middleton, 1986a; Goudie and Middleton, 1992; Qian et al., 2002; Engelstaedter et al., 2003). While this method is not affected by the uneven observation schedule at different stations, any dust reports within a day are treated as a single event, regardless of the duration and intensity. An alternative method is to treat each reported dust code as an individual event, and derive the dust event frequency as:

$$FR = \frac{N_{du}}{N_{ww}} \times 100\% \tag{1}$$

where $N_{du}$ is the number of dust reports, and $N_{ww}$ is the total number of valid (i.e., non-missing) $ww$ reports (Kurosaki and Mikami, 2003; Shao et al., 2013). To exclude the possible influence of fog events, any $ww$ observations with relative humidity exceeding 90% are treated as non-dust, even if a dust code is reported. Eq. 1 can be applied to each dust code separately or in groups of dust codes describing similar weather types (Kurosaki and Mikami, 2005; Camino et al., 2015). In addition, dust codes can be weighted to calculate a composite dustiness index based on the their relationship with dust mass concentration and visibility (O'Loingsigh et al., 2014). In general, $FR$ and its variants are limited by the qualitative nature of $ww$ observations in quantifying the dust burden.

## 2.2  Visibility

Visibility, also referred to as meteorological optical range, is defined as the greatest distance at which a dark object of suitable dimensions can be seen and recognized against the horizon by unaided human eyes (WMO, 2018). Visibility was traditionally determined by trained observers at ground level viewing reference objects at known distances from the station, but has been increasingly relying on automated systems since the 1990s. Visibility is typically reported at an increment of 100 m below 5 km, 1 km for 6–30 km, and 5 km for 35–70 km (WMO, 2018).





Compared to $ww$, visibility provides more quantitative information of the dust activity. The relationship between visibility ($V$, km) and the ambient dust burden can be described as:

$$Y = a \times V^b \tag{2}$$

The dependent variable $Y$ can take the form of volume extinction coefficient of the atmosphere (km$^{-1}$) or the ambient dust mass concentration (μg m$^{-3}$). In the former case, Eq. 2 is the Koschmieder equation with $b = -1$ and $a = 1.5$–$3.9$, depending on the contrast of visible objects against horizon sky and the critical contrast threshold (2–5%) of human eyes (Li et al., 2016). In the latter case, numerous studies showed that visibility can be used to predict the dust mass concentration with reasonable accuracy (Chepil and Woodruff, 1957; Mohamed et al., 1992; Shao et al., 2003; Baddock et al., 2014; Jugder et al., 2014; Camino et al.,
2015). These studies suggest that $a$ is primarily linked to the dust particle size limit (e.g., 2.5 or 10 μm), while $b$ varies from $-1.1$ to $-0.5$, depending on the distance of the station from the source, and hence the dust particle size distribution.

Given the inverse relationship between visibility and dust burden, harmonic mean, rather than arithmetic mean, should be used to calculate the average of visibility data. In fact, according to the Koschmieder equation, the harmonic mean visibility can be viewed as an equivalent of the extinction coefficient. As shown in Table 1, the harmonic mean visibility can be used
as a quantitative reference to distinguish the dust event categories. Here, in addition to $FR$, a visibility-based measure of dust burden is defined as:

$$VI = \frac{1}{V} \times FR \tag{3}$$

In this case, $V$ is the harmonic mean visibility associated with dust events. The rationale of $VI$ is to combine the qualitative $ww$ and quantitative visibility observations to account for both the frequency and varying intensity of dust events. As shown
later, $VI$ allows more accurate tracking of the dust burden than $FR$.

### 2.3   Other data

To test if surface weather stations provide a consistent view of dust variability, $FR$ and $VI$ are compared with a MODIS-derived dust climatology and two global aerosol reanalysis products, including the Atmospheric Composition Reanalysis 4 from the Copernicus Atmosphere Monitoring Service (CAMS) and Modern-Era Retrospective Analysis for Research and Applications
version 2 (MERRA2) from the NASA Global Modeling and Assimilation Office. Voss and Evan (2020) recently developed a global dust optical depth (DOD) climatology for 2001–2018 using MODIS level-3 daily AOD, which is a composite of quality-assured aerosol retrieval over ocean and land based on the Dark Target and Deep Blue algorithms, respectively (Hsu et al., 2013; Levy et al., 2013). To separate the dust component over ocean, Voss and Evan (2020) followed Kaufman et al. (2005) by removing the contribution of fine-mode (anthropogenic and biomass burning) and sea salt aerosols from the total
AOD. Over land, Voss and Evan (2020) used the method of Ginoux et al. (2012) to separate the dust signal based on the unique spectral signature of dust Ångström exponent and single scattering albedo. CAMS and MERRA2 represent recent advances in developing atmospheric composition reanalysis using global model systems with capabilities to assimilate satellite observations of atmospheric aerosols and gaseous species (Gelaro et al., 2017; Inness et al., 2019). CAMS is produced by





the ECMWF Integrated Forecast System (IFS) which assimilates bias-corrected total AOD from AATSR (Advanced Along-
Track Scanning Radiometer) and MODIS from October 2002 onward. MERRA2 is produced by the Goddard Earth Observing
System version 5 (GEOS-5) which assimilates multiple aerosol products, including the AVHRR AOD over ocean prior to 2002,
MODIS AOD over ocean, MISR AOD over land (including deserts), as well as ground-based AOD retrieval from AERONET
(Randles et al., 2017).

   To investigate the climate drivers of dust changes, two physical variables are examined through empirical analysis: surface
wind speed measured by weather stations, and soil moisture from the European Space Agency Climate Change Initiative (CCI)
v04.7 combined product. The CCI soil moisture is derived from a list of historical and current passive and active microwave in-
struments, and is available as a global daily gridded product from 1978 to 2019 (Dorigo et al., 2017; Gruber et al., 2017, 2019).
Microwave sensors are highly sensitive to the water content in the top soil layer, which inhibits dust emission by increasing the
bonding force of soil particles and supporting vegetation growth. Several standard climate indices are also used, including the
self-calibrating Palmer Drought Severity Index (scPDSI), North Atlantic Oscillation (NAO), Atlantic Multidecadal Oscillation
(AMO), Pacific Decadal Oscillation (PDO), and Arctic Oscillation (AO). scPDSI is a standardized index used to track long-
term changes of drought across different climate zones, and calculated from the CRU TS v4.03 global monthly precipitation
and Penman-Monteith potential evapotranspiration data (Van Der Schrier et al., 2013). The scPDSI data are obtained from
https://crudata.uea.ac.uk/cru/data/drought/, and other climate indices from https://psl.noaa.gov/data/climateindices/.

**3   Data quality assessment**

**3.1   Continuity of present weather reports**

Currently, ISD includes nearly 30,000 stations (active and inactive) worldwide. As part of the initial data screening, stations
without geolocation information are excluded. Only the stations within 50°S–70°N are considered. Given the purpose of this
study, only the stations with at least 1 dust event report (including from construction, traffic, or even false recording) between
1940 and 2019 are included. Furthermore, for a station to be included in any given year, two conditions must be met: (a) the
station must have at least 1 $ww$ report per day, and (b) $ww$ must be reported in every month of the year. These rules are intended
to exclude the stations with uneven sampling on a yearly basis.

   After the above screening, there are a total of 13,501 stations from various report types (e.g., SYNOP, METAR) and platforms
(e.g., fixed land-based, mobile land-based, ship, buoy). The geographic distribution of these stations is shown in Fig. S1. While
these stations provide homogeneous data coverage on the yearly basis, not all of them are suitable for long-term analysis. As
a station is added, retired, moves to a new location, upgrades the instruments, or changes the reporting method (e.g., from
manual to automatic), it is likely to cause discontinuity in $N_{ww}$ and consequently, $FR$ and $VI$. Figure 1a shows that $N_{ww}$
varies significantly from year to year, primarily due to changes in the number of stations. Due to a change in the WMO Global
Telecommunication System, a drastic increase in the number of stations in 1973, from 570 to 7,275, led to 18 times increase
in $N_{ww}$ (i.e., from 1.1 million to 20.8 million) and 30 times increase in $N_{du}$ (i.e., from 7,412 to 220,840). As a result, ISD



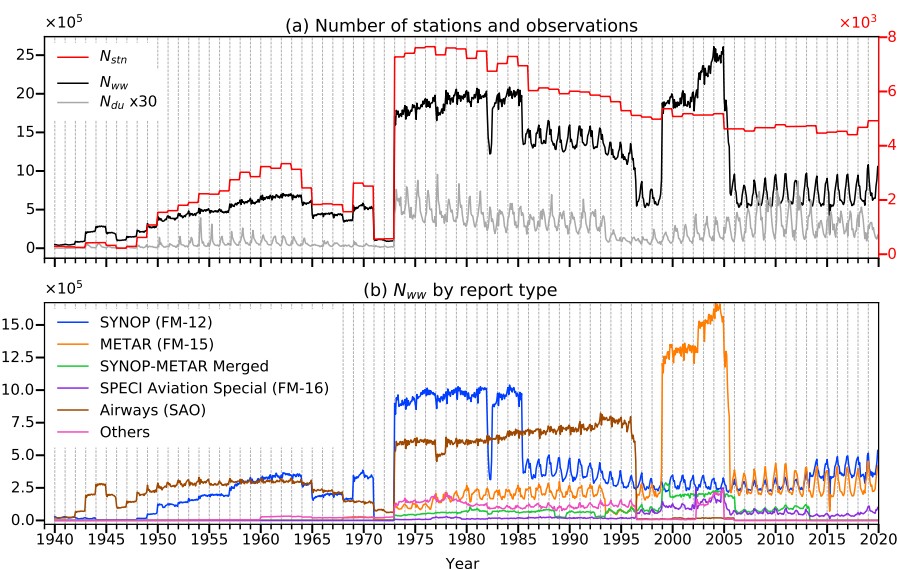

**Figure 1.** Analysis of the continuity of present weather ($ww$) reports in ISD. (a) Global monthly number of stations ($N_{stn}$), number of $ww$ reports ($N_{ww}$), and number of dust event reports ($N_{du}$). For clarity $N_{du}$ is multiplied by 30; (b) $N_{ww}$ from different report types.

provides nearly global coverage of the world's prominent dust sources starting from 1973, which can also be seen from the enhanced seasonality of $N_{du}$ after 1973. At local level, however, the dust record may extend back to the 1940s at some stations.

After 1973, $N_{ww}$ shows multiple step changes in 1982, 1985, 1996, 1999, 2002, and 2005. While the $N_{ww}$ discontinuity can be partly explained by the number of stations, other factors are also responsible, such as the observation schedule or method
(e.g., from manual to automated). For instance, the $N_{ww}$ increase in 2002 is likely due to more frequent observations (including scheduled and unscheduled), since the number of stations remains the same. In contrast to $N_{ww}$, $N_{du}$ shows little variations after 1973, except for a noticeable decline between 1993 and 2001, which may result from the reduced $ww$ observations. In addition, both $N_{ww}$ and $N_{du}$ vary annually: more frequent $ww$ observations are made during boreal winter months (Nov–Feb), while dust events are more frequent in spring and early summer (Feb–Jun).

To understand the underlying causes of $N_{ww}$ discontinuity, Fig. 1b shows the monthly $N_{ww}$ by the report type or station network of $ww$ observations. The most important report types are SYNOP, METAR, Airways, and SPECI. SYNOP, called FM-12 by WMO, is an international standard code format for reporting weather observations from fixed manned and automatic stations typically at 6- or 3-hourly intervals. METAR (FM-15), or Meteorological Terminal Aviation Routine Weather Report, is an international standard code for hourly aviation weather observations. In addition to routine observations, METAR also
includes SPECI (Aviation Selected Special Weather Report) for unscheduled reports of rapid weather change of significance to aviation. In addition, ISD includes hourly Surface Airways Observations (SAO) that extend back to the time when commercial aviation began in the United States. The SAO code format was later replaced by METAR, which explains the sudden cessation of SAO record in 1996. It is clear from Fig. 1b that the discontinuity in $N_{ww}$ results from the inhomogeneous coverage by the





**Table 2.** Trends of the harmonic mean visibility (km yr$^{-1}$) associated with dust weather codes ($ww$ = 06–09, 30–35, 98). Only significant trends at 95% confidence level are shown. Dust weather codes are described in Table 1.

| $ww$ code | 1973–2019 | | 1986–2019 | |
| --- | --- | --- | --- | --- |
| | All report types | SYNOP only | All report types | SYNOP only |
| 06 | −0.07 | - | −0.06 | 0.02 |
| 07 | −0.04 | - | −0.04 | - |
| 08 | 0.09 | 0.12 | - | - |
| 09 | 0.09 | 0.06 | 0.1 | 0.06 |
| 30 | −0.02 | - | −0.02 | −0.02 |
| 31 | −0.02 | - | −0.01 | 0.02 |
| 32 | - | - | - | 0.01 |
| 33 | −0.01 | −0.01 | - | - |
| 34 | −0.003 | - | - | - |
| 35 | - | - | - | - |
| 98 | 0.01 | 0.03 | - | - |

diverse report types in ISD, which have different station coverage and observation schedules. In general, SYNOP provides the

most continuous $ww$ record since 1986. Also, SYNOP provides fairly sufficient coverage of global dust sources, except for the United States where $ww$ is primarily reported in METAR and SAO (see Fig. S2). Therefore, the SYNOP data between 1986 and 2019 are most suitable for global-scale dust analysis.

## 3.2 Continuity of dust event reports

Since $N_{du}$ consists of reports of 11 $ww$ codes, it is necessary to take a critical look at the continuity of individual dust codes

and their utility for tracking dust changes. Changes in dust code definitions or reporting guidelines may cause a discontinuity in the code usage and quantification of the dust burden. Any significant change in the dust code usage, if unexplained by the $ww$ schedule, is a possible sign of discontinuity. Figure 2 shows that among the dust codes, $ww$ = 09 shows persistently low usage prior to a three-fold increase in 2014. Similar behaviors are not found in other codes. O'Loingsigh et al. (2014) found that a code definition change resulted in significant decrease of $ww$ = 09 usage after 1973 in Australia. The implication is that

using $ww$ = 09 by itself could lead to false detection of the dust trend. Statistically, $ww$ = 09 accounts for only 1% of all dust reports, and therefore has negligible influence when all dust codes are combined together. Unlike $ww$ = 09, the abrupt increase of $ww$ = 34–35 in 2001 after persistently low usage is most likely due to intensified dust activity, which is also observed in $ww$ = 30–33. Interestingly, $ww$ = 06–07, 09, 30–35 show coherent seasonal cycles, suggesting that dust in suspension ($ww$ = 06), blowing dust ($ww$ = 07), and dust storms ($ww$ = 09, 30–35) are governed by similar physical processes. In contrast, dust devils

and swirls ($ww$ = 08) and thunderstorm-accompanied dust events ($ww$ = 98) display different seasonal cycles.





**Figure 2.** Analysis of the continuity of dust weather code usage. From top to bottom: monthly number of reports of all and individual dust codes in the global SYNOP data. Horizontal lines are all-time averages. Dust weather codes are described in Table 1.





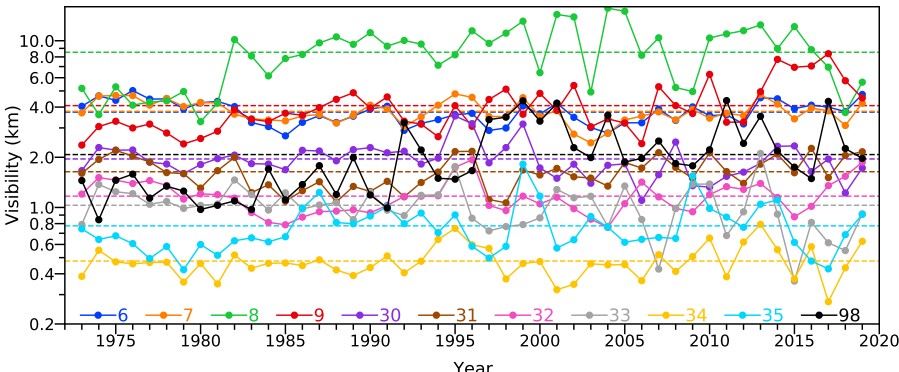

**Figure 3.** Temporal consistency of the harmonic mean visibility associated with dust weather codes. Dash lines are all-time averages. Dust weather codes are described in Table 1.

According to Table 1, dust codes differ from each other by the event intensity, as well as the location and timing of occurrence. Figure 3 shows that while there are significant year-to-year fluctuations in the harmonic mean visibility associated with dust codes, they generally fall into three clusters: $ww$ = 06–09 (3.7 km), $ww$ = 30–32, 98 (1.5 km), and $ww$ = 33–35 (0.7 km). Due to the qualitative nature of dust event reports, systematic changes over time in the dust burden cannot be fully

captured by the categorical dust event reports and the derived $FR$, which implicitly assumes that the dust codes provide a consistent representation of dustiness over time. Here the assumption is tested by examining if there are systematic changes in the harmonic mean visibility (equivalent to extinction coefficient) of each dust code. Table 2 summarizes the trends (km yr$^{-1}$) of annual harmonic mean visibility associated with the dust codes, based on four subsets of ISD. The trends are calculated using the non-parametric Mann-Kendall test associated with Theil-Sen's slope (Hussain and Mahmud, 2019). The trends should be

interpreted with the fact that the dust codes differ greatly in the mean visibility, and that $ww$ = 06–07 account for the majority of dust events (93%). For example, assuming the visibility of $ww$ = 06 increases from the all-time mean of 3.7 to 3.9 km at a rate of 0.02 km yr$^{-1}$ (Table 2), it is equivalent to 5% decrease in the extinction coefficient after 10 years. In contrast, for $ww$ = 31, assuming its visibility increases from the all-time mean of 1.6 to 1.8 km at the same rate, it is equivalent to a 12.5% reduction in the extinction coefficient. This simplified example demonstrates that due to systematic changes in the dust codes,

variations in the $FR$ should be interpreted with caution. Since $ww$ = 06–07 account for the majority of dust reports, it is most important to ensure that $ww$ = 06–07 are subject to minimal inconsistency. It appears that the SYNOP data since 1986 meet the requirements on the dust code continuity and $ww$-visibility consistency for global analysis.



**Figure 4.** Decadal mean dust event frequency ($FR$). Gray dots are stations with $FR < 1\%$.

## 4   Global dust pattern and trend

### 4.1   Decadal mean pattern

Using Eq. 1, the decadal mean $FR$ is calculated for stations with at least 5 years' data in each decade, as shown in Fig. 4. The $FR$ map provides a consistent view of present-day dust source distribution and transport pathway compared to satellite remote sensing (Prospero et al., 2002; Ginoux et al., 2012). Specifically, the most active sources associated with $FR > 40\%$ are located





**Figure 5.** Same as Fig. 4 but for $VI$. Gray dots are stations with $VI < 0.01$ km$^{-1}$.

in North Africa (Sahara), Middle East (Arabian Desert), East Asia (Taklamakan), and South Asia (Thar). Some of these sources are responsible for the transcontinental export of aeolian dust with global consequences on the climate and air quality (e.g.,
Yu et al., 2012, 2015). Comparatively less active sources ($FR = 10$–$40\%$), ranked by $FR$, include the Gobi covering much of northern China and southern Mongolia, Patagonia Desert in South America, the sandy and solonchak deserts (e.g., KaraKum, Aralkum) in Central Asia, the remote interior of Australia, Chihuahuan Desert in northern Mexico and southwestern USA, Kalahari Desert in southern Africa, and the Icelandic deserts. Several areas without local sources, including the Caribbean,





Iberian Peninsula, Korea Peninsula and Japan, also experience frequent dust weather due to the outflow from African and
Asian sources.

The distinctive changes of decadal mean $FR$ can be linked to multidecadal climate variations, especially the occurrence of
mega-drought events lasting several years or even decades. Elevated dust activity can be observed in areas affected by persistent
drought, where the reduction of soil moisture and vegetation leaves the exposed, dry soil prone to wind erosion. For example,
a striking feature in the 1950s is the widespread, frequent dust events in the U.S. Southwest and Midwest, with several stations
reporting $FR > 20\%$ in the High Plains of Texas and Colorado. The heightened dust activity was fueled by a 11-year-long
(1946–1956) drought that afflicted a massive area centered in the Southwest U.S. (Fye et al., 2003). The 1950s drought was
characterized by a prolonged lack of precipitation and excessive warm temperatures, which caused crop failure and livestock
feed shortage (Goudie and Middleton, 1992). As the drought came to an end in the spring of 1957, $FR$ started to decline and
has since remained low in the last 50 years.

Similarly, North Africa experienced progressively drier conditions during the 1970–80s in the Sahel, a semiarid dryland belt
at the southern border of Sahara Desert (Giannini et al., 2008). The Sahelian drought was triggered by anomalous sea surface
temperature (SST) in the tropic Atlantic and Indian Ocean (Dai, 2011). The Sahelian dust frequency during drier-than-normal
years, especially in the 1980s when drought was most severe, is significantly higher compared to the pre- or post-drought
periods. The drought-induced dust enhancement is also evident from the frequent dust weather observed downstream, including
the Caribbean, Gulf of Mexico, and Iberian Peninsula. This is consistent with the long-term in situ dust measurements in
Barbados and Miami, Florida, indicating a positive correlation between the Sahel dry anomaly and African dust outflow across
the tropical North Atlantic (Prospero and Lamb, 2003; Zuidema et al., 2019). With the amelioration of Sahelian drought in the
2000s, $FR$ experienced significant decreases at the source and downwind, consistent with ground and satellite observations
(Hsu et al., 2012; Li et al., 2014). In the past decade, increased dust activity can be observed in West Africa and the Middle
East, which will be discussed later.

Figure 5 shows very similar pattern in the decadal mean $VI$. A cutoff value of 0.01 km$^{-1}$ is used, equivalent to the extinction
coefficient of a molecular atmosphere (in the absence of aerosols and hydrometers). In other words, Fig. 5 highlights the
stations where the dust contribution to visibility reduction exceeds the capacity of a pure molecular atmosphere. It appears that
stations near dust source areas stand out clearly, since these stations tend to experience more frequent severe dust events with
disproportionately low visibility values.

## 4.2   Interannual variability and trend

Figure 6 shows the global monthly $FR$ and $VI$ derived from the SYNOP data between 1986 and 2019, which provide the most
homogeneous and continuous global dust record (Sect. 3). $FR$ and $VI$ have very similar variations ($r = 0.91$), both reaching a
maximum in 1988 and a minimum in 1996/97. At annual scale, both $FR$ and $VI$ peak in the spring (March-April-May). To test
if weather stations provide a consistent view of global dust variations, $FR$ and $VI$ are compared with the datasets described in
Sect. 2.3. Figure 7 reveals systematic differences between the DOD from MODIS, CAMS, and MERRA2. For the overlapping
period (2003–2018), the mean DOD is 0.042, 0.013 and 0.024 from MODIS, CAMS and MERRA2, respectively. CAMS also





**Table 3.** Pearson correlation coefficients ($r$) between weather station-based dust records ($FR$ and $VI$) and dust datasets from satellite observations and model reanalysis, based on deseasonalized monthly anomalies for the overlapping periods. For all $r$ values, $p < 0.001$.

|  |  | $FR$ | $VI$ |
|---|---|---|---|
| MODIS[a] | Dust optical depth | 0.51 | 0.62 |
| CAMS[b] | Dust optical depth | 0.57 | 0.64 |
|  | Column mass density | 0.53 | 0.56 |
| MERRA2[c] | Emission flux | 0.22 | 0.36 |
|  | Dust optical depth | 0.35 | 0.37 |
|  | Column mass density | 0.35 | 0.37 |
|  | Surface mass Concentration | 0.42 | 0.48 |

[a] 2001–2018; [b] 2003–2019; [c] 1986–2019

**Table 4.** Summary of global annual dust trends. Only significant trends at 95% confidence level are shown.

|  | 1986–2019 | 1986–1996 | 1996–2019 | 2001–2018 |
|---|---|---|---|---|
| $FR$ (% yr$^{-1}$) | $2.0\times10^{-2}$ | $-2.3\times10^{-1}$ | $8.5\times10^{-2}$ | $1.0\times10^{-1}$ |
| $VI$ (km$^{-1}$ yr$^{-1}$) | - | $-8.0\times10^{-4}$ | $1.9\times10^{-4}$ | $1.9\times10^{-4}$ |
| MERRA2 (DOD yr$^{-1}$) | $1.4\times10^{-4}$ | $-3.6\times10^{-4}$ [a] | $1.8\times10^{-4}$ [b] | - |
| MODIS (DOD yr$^{-1}$) |  |  |  | $1.2\times10^{-4}$ |
| CAMS (DOD yr$^{-1}$) |  |  |  | - [c] |

[a] 1986–1992; [b] 1992–2019; [c] 2003–2019

appears to be less variable than MODIS and MERRA2. The offsets in the mean DOD can be attributed to a number of factors. For example, MODIS DOD is based on once-per-day observations over both ocean and land, including deserts where large AOD values (up to 5) are allowed in the Deep Blue algorithm, whereas CAMS and MERRA2 DOD are generated by global models with assimilation of quality-controlled total AOD over ocean and dense vegetation. The lack of AOD assimilation over dust source areas may be the primary reason why the CAMS and MERRA2 DOD are lower. The global models used to produce CAMS (ECMWF IFS) and MERRA2 (NASA GEOS-5) have a variety of differences in the model configuration, dust parameterization, dust optical properties, meteorological input, and AOD assimilation, all of which may contribute to the discrepancy between CAMS and MERRA2 DOD.

By subtracting the climatological monthly mean from the original data for the overlapping periods, Pearson's correlation is performed on the deseasonalized monthly anomalies of all dust variables, which are summarized in Table 3. In general, surface weather stations show consistent year-to-year dust variations with MODIS and model reanalysis. Interestingly, $VI$ has a higher correlation with all dust quantities than $FR$ does, which confirms that the combination of $ww$ with visibility can better quantify

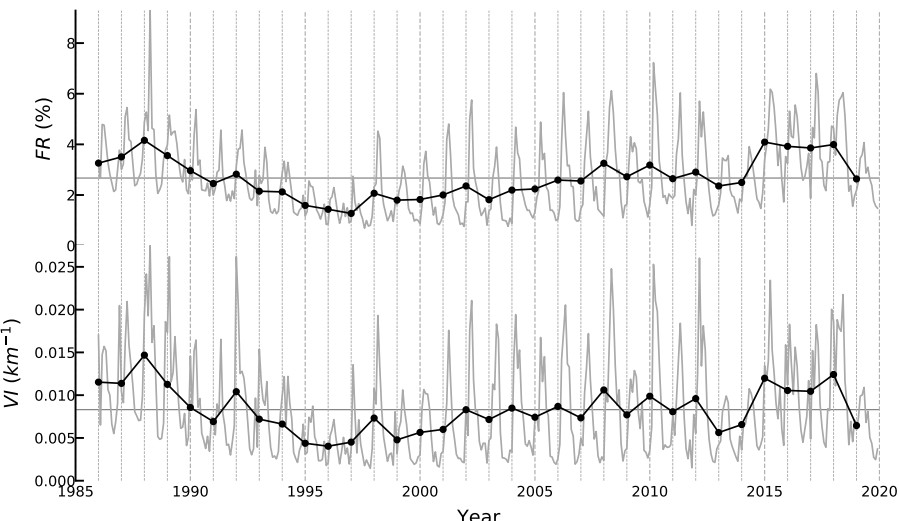

**Figure 6.** Global monthly (gray) and annual (black dotted) $FR$ and $VI$. Horizontal lines are all-time averages.

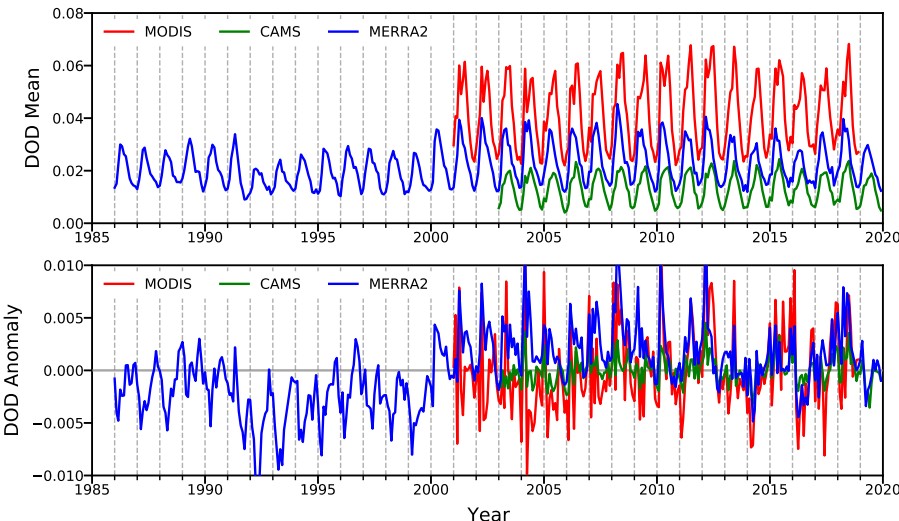

**Figure 7.** Comparison of global monthly means and anomalies of dust optical depth (DOD) from MODIS, CAMS, and MERRA2.

the dust burden than $ww$ alone. Also, comparison with MERRA2 reveals that both $FR$ and $VI$ have a higher correlation with the surface mass concentration than with the column properties (DOD and column mass density), which is consistent with the fact that weather stations are more representative of near-surface dust conditions. At annual scale, Fig. 8 shows some mismatch in the dust seasonality: MODIS and CAMS show maximum DOD in the summer while the MERRA2 DOD peaks in the spring, in closer agreement with $FR$ and $VI$. The seasonality mismatch may be caused by inconsistent sampling and/or differences

in the physical measurements. For example, due to logistical issues, weather stations are not uniformly distributed and located





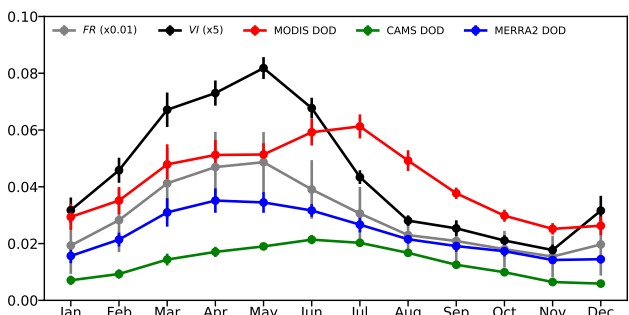

**Figure 8.** Annual cycle of global mean $FR$, $VI$, and dust optical depth (DOD) averaged over 2003–2018. For clarity $FR$ (%) and $VI$ (km$^{-1}$) are multiplied by 0.01 and 5, respectively.

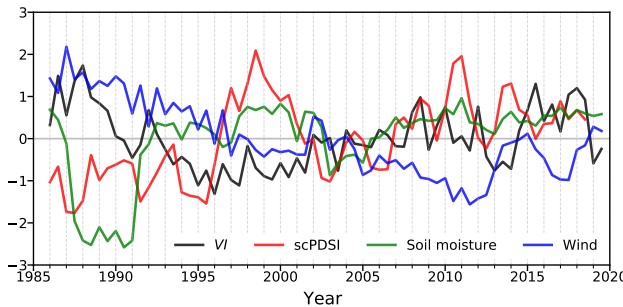

**Figure 9.** Covariation of global mean $VI$, self-calibrating Palmer Drought Severity Index (scPDSI), ESA CCI soil moisture, and surface wind speed. For clarity 6-month running averages of standardized anomalies are shown.

close to human settlements. While weather stations do not cover the interior of deserts, they may still be advantageous over passive remote sensors in capturing the dust variations near the source areas, where the high surface reflectance of exposed, dry soils dominates the light backscatter in visible wavelengths. Weather stations also have more frequent observations on a daily basis, whereas polar-orbiting instruments (such as MODIS) usually make a single overpass each day. In addition, weather

stations focus on dust conditions in the lower boundary layer, whereas satellites retrieve the total aerosol column and are unable to detect aerosols below clouds.

Trends are further calculated from the monthly anomalies using the pyMannKendall package developed by Hussain and Mahmud (2019), which consists of multiple Mann-Kendall test options to accommodate the seasonality and serial correlation in the data. The Mann-Kendall test is a non-parametric test of the presence of monotonic trend in the data, and has advantage

over parametric methods (e.g., t test) for its insensitivity to outliers, missing values, and the statistical distribution of the data. The Mann-Kendall test is designed for serially independent data and thus can be influenced by the presence of autocorrelation in the data, which either increases the uncertainty of estimated trends or prolongs the length of time period required to detect a given trend (Weatherhead et al., 1998). It is found that the $FR$ and $VI$ anomalies exhibit strong autocorrelation, with a correlation coefficient of 0.7 and 0.5 at 1 month's lag, respectively. Significant autocorrelation is also found in the DOD

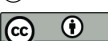



datasets. To eliminate the influence of serial correlation on the trend test, Yue et al. (2002) suggested a pre-whitening method in which the AR(1) component is removed from time series after first removing the linear trend, which is later added back to the data. This method is applied to detect the presence of monotonic trends, while the magnitude of trends is estimated by the Theil-Sen's slope, defined as the median slope of all pairs of ordered data points.

The global dust trends are summarized in Table 4. Previously, Shao et al. (2013) found a negative global trend of visibility-
derived dust mass concentration from 1984 to 2012. Similar negative trends are also found in $FR$ and $VI$ during the same time period. For the period 1986–2019, however, both $FR$ and MERRA2 DOD show positive trends, while $VI$ suggests no significant change. Thus, it appears that the overall trend has been reversed due to the enhanced dust activity in recent years. Figure 6 shows a decline in global dust from 1985 to 1996/97, followed by an upward trend. Figure 7 shows a similar non-linear behavior in the MERRA2 DOD which, however, reaches a minimum in 1992, or 4 years earlier than the weather stations
suggest. All datasets suggest that the positive dust trend continues into the past two decades (2001–2018), except for MERRA2 and CAMS DOD which show no significant trends due to the strong autocorrelation in the data.

The decadal mean $FR$ and $VI$ (Fig. 4 and 5) indicate mega-drought events are associated with extremely active dust periods in the 20th century. To investigate the dust-drought connection on the interannual timescale, Fig. 9 shows the normalized monthly anomalies of global mean $FR$, $VI$, scPDSI, soil moisture, and wind speed. Both scPDSI and soil moisture are calcu-
lated from gridded products over the land area between 50°S–70°N, while the wind speed is from weather stations. Negative scPDSI values indicate dry conditions, while positive values indicate wet conditions. Globally, soil moisture is correlated with scPDSI ($r = -0.45$, $p < 0.001$), both showing severe dry anomalies in the late 1980s and early 1990s. The global mean $VI$ is negatively correlated with scPDSI ($r = -0.17$, $p < 0.001$) and soil moisture ($r = -0.21$, $p < 0.001$). Moreover, a maximum correlation ($r = -0.3$, $p < 0.001$) is found between $VI$ and soil moisture, when soil moisture leads by 14 months. It implies that
soil moisture exerts a memory effect on the dust activity, which could be explained, at least partly, by the role of vegetation. The hypothesis is that a wetter-than-normal year is likely to be followed by weak dust activity in the next year, either by live plants or dead materials from the previous growing season, which not only reduces the exposed soil surfaces, but also increases the surface roughness and slows down the surface winds. This effect is most prominent in semiarid areas (e.g., temperate grasslands, shrublands) where dust emission is highly sensitive to precipitation and vegetation (Shinoda et al., 2011).

Figure 9 shows a steady decline of the global mean wind from 1986 to 2011, followed by a fast recovery in the past decade. The wind stilling and recent strengthening have been widely reported (e.g., Vautard et al., 2010; McVicar et al., 2012; Zeng et al., 2019). The global mean $VI$ shows a moderate correlation with the wind speed ($r = 0.19$, $p < 0.001$). Since the majority of weather stations are distant from dust sources and impacted by transport events only (which explains why $ww = 06$ is the most frequently reported), the global mean wind may not accurately represent the wind power relevant for dust uplifting, leading
to the lower-than-expected dust-wind correlation. This is supported by the fact that the global mean wind is inversely related to soil moisture ($r = -0.49$, $p < 0.001$), which can be explained by the wind effect on terrestrial evapotranspiration that is dominant in humid areas (McVicar et al., 2012).





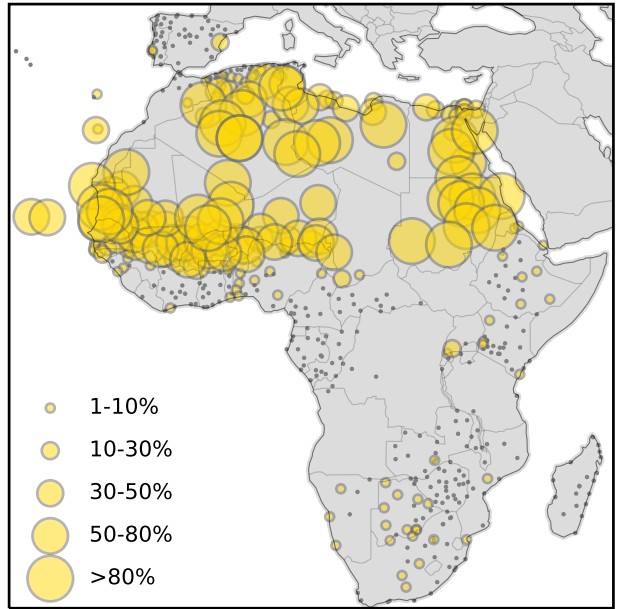

**Figure 10.** Dust event frequency ($FR$) in Africa based on the SYNOP present weather report between 1986 and 2019. Gray dots are stations with $FR < 1\%$.

## 5 Regional dust variations and trends

### 5.1 North Africa

Figure 10 shows the $FR$ in Africa by averaging the SYNOP data between 1986 and 2019. All stations from the SYNOP network are included, regardless of the data length of individual stations. As shown in the top panel of Fig. 11, $N_{ww}$ fluctuates from year to year, but has no significant step changes. The stability of $N_{ww}$ can be improved by keeping only the stations with a minimum of record length (e.g., 17 years or 50% of 1986–2019). The station screening is found to have minor effects on the qualitative behaviors of regional mean $FR$ and $VI$.

African dust emission and transport respond to a number of meteorological phenomena characterized by multi-scale variability in the atmospheric circulation and precipitation. Unlike Shao et al. (2013), no significant correlation is found between either $FR$ or $VI$ and AMO, which describes a spatially coherent pattern of oscillatory changes in the North Atlantic SST and has been suggested as a driver of low frequency variability of Sahel rainfall (Wang et al., 2012). Nonetheless, the dust-AMO correlation becomes statistically significant ($r = -0.2$, $p < 0.05$) when the AMO index leads by 7 months. Also, In contrast to

Shao et al. (2013) who found no correlation between the visibility-derived dust concentration and NAO, both $FR$ and $VI$ are positively correlated ($r = 0.22$, $p < 0.001$) with the Jones NAO index (Jones et al., 1997) (Fig. 11). A similar relationship is found between NAO and the MERRA2 DOD ($r = 0.28$, $p < 0.001$). NAO consists of a north-south dipole of pressure anomalies with action centers near Iceland and over the subtropic Atlantic, and affects African dust by controlling the strength and



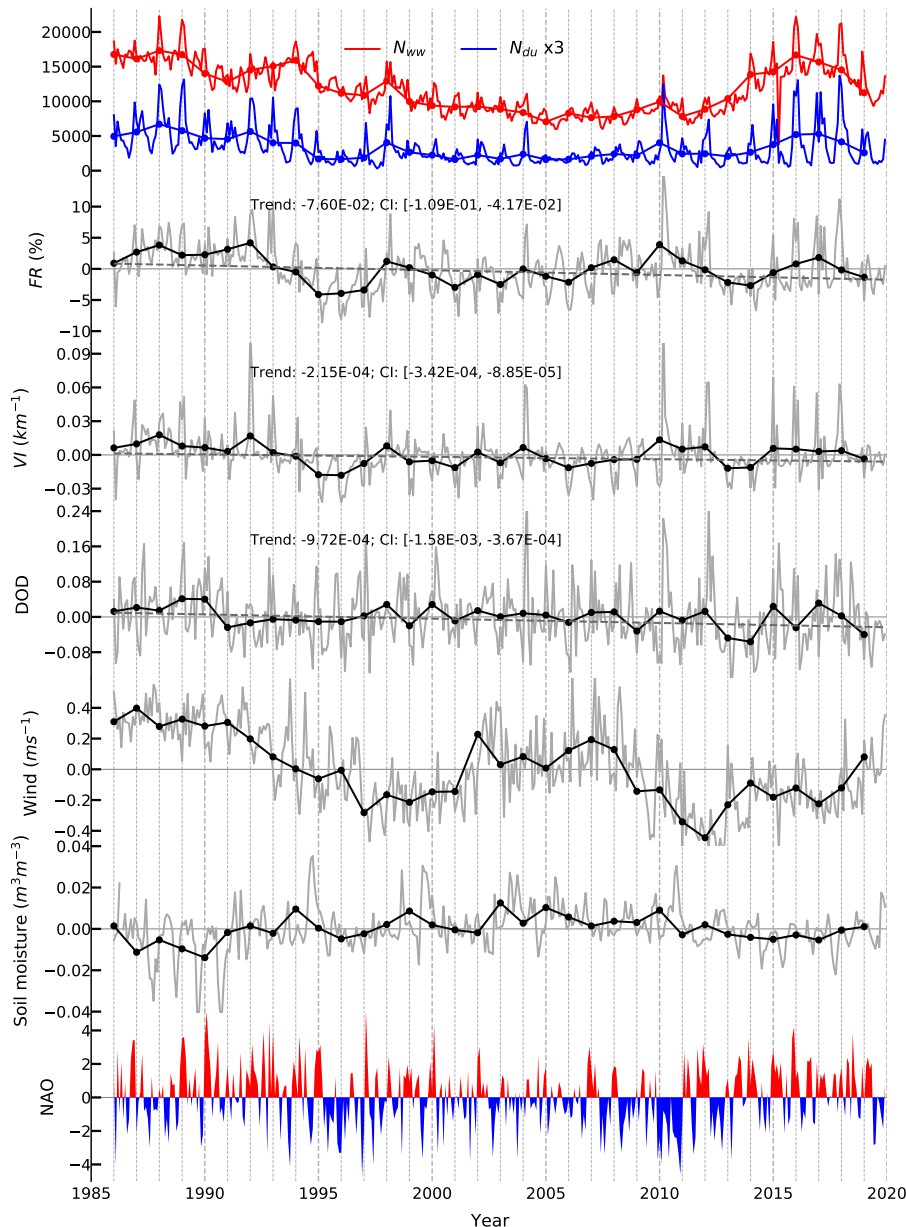

**Figure 11.** Dust variation in North Africa from 1986 to 2019. From top to bottom: monthly $N_{ww}$ and $N_{du}$, and anomalies of $FR$, $VI$, MERRA2 DOD, wind speed, soil moisture, and the Jones North Atlantic Oscillation (NAO) index. DOD and soil moisture are averaged over region 10°N–20°N, 20°W–20°E; Dotted curves are annual averages. Dashed lines are Theil-Sen linear regression of $FR$, $VI$ and DOD, with the annual trends and confidence intervals (CI) shown.

direction of the North Atlantic westerlies and storm track, which are responsible for much of the heat and moisture transport
to Europe and North Africa (Hurrell, 1995). Strong positive NAO phases yield stronger westerlies and increased precipitation



in mid- and high-latitudes in Europe, while the opposite pattern is associated with increased storm activity and precipitation in southern Europe and North Africa (Moulin et al., 1997). Satellite records from recent decades suggested that NAO is strongly correlated with African dust outflow to the North Atlantic and Mediterranean (Chiapello and Moulin, 2002; Chiapello et al., 2005). While NAO plays an important role in modulating the African dust activity, NAO may also interact with or covary with

other teleconnection patterns and local centers of action, with consequences on the wind and rainfall regimes in North Africa (Evan et al., 2016).

Historical NAO data reveal that recurring positive NAO phases dominate an extended period of 1980–1995, followed by a transition to the strong negative NAO phase in the winter of 1995/96 (Halpert and Bell, 1997). This climate shift has widespread impact on the temperature and precipitation pattern across the North Atlantic, Eurasia, and northern Africa, and consequently,

on the African dust emission and transport. Consistent with previous studies, Fig. 11 shows that African dust was at a historic high during the late 1980s due to abnormally strong winds and low soil moisture, both favorable for dust production (e.g., Prospero and Lamb, 2003; Chiapello et al., 2005; Evan et al., 2016). The dust peak was followed by a declining trend, as shown by $FR$, $VI$ and MERRA2 DOD consistently, which was suggested to be primarily driven by the weakening of surface winds (Ridley et al., 2014; Chin et al., 2014). Indeed, Fig. 11 reveals a steady decrease of wind speed from 1986 to 1997,

followed by a temporary recovery between 2002 and 2008. The wind stilling from 1986 to 2000 was also accompanied by an increase of soil moisture, which could further contribute to the dust suppression. Particularly, the soil moisture shifted from below to above average, following the NAO transition from recurring positive to strong negative phases in the winter of 1995/96. As the soil remained abnormally wet during 2002–2008, dust continued to be suppressed (i.e., both $FR$ and $VI$ remained below average), even if the wind speed returned to above average. In the past decade (2011–2019), as NAO returned

to recurring positive phases in early spring 2011, there has been a drastic increase of dust activity in North Africa, which can be explained by a decrease of soil moisture to below average and an increase of wind speed.

## 5.2  Middle East

Middle East is arguably the second largest contributor after North Africa to global dust emission on an annual basis. Earlier weather station records indicate that the region's dust outbreak is largely due to the passage of Mediterranean cyclones in the

spring and the northeasterly Shamal wind during the summer, especially over the Tigris-Euphrates river basin where the alluvial deposits are a rich source of dust uplifting (Middleton, 1986b). As shown in Figure 12, There is a cluster of stations with $FR$ > 50% in the lower Mesopotamian Plain (Iraq), where the fine-grained alluvium becomes highly susceptible to erosion under drought conditions. Other local sources that can be identified from the $FR$ map include the Rub' al Khali, Ad Dahna, and An Nafud sandy deserts. Dust outflow from Sahara was also suggested as a major source of dust weather in the northern coast of

Red Sea (Notaro et al., 2013).

Figure 13 shows that $N_{ww}$ increases over time as more stations are ingested into ISD. $N_{du}$ shows synchronous changes as $N_{ww}$, indicative of continuous observation schedule in the SYNOP data. Even though $FR$, $VI$ and MERRA2 DOD consistently show a positive trend between 1986 and 2019, the period is characterized by a nonlinear dust behavior, with a negative trend for 1986–1998 and a positive trend for 1998–2019. The initial decline was likely due to the combined effect of reduced wind





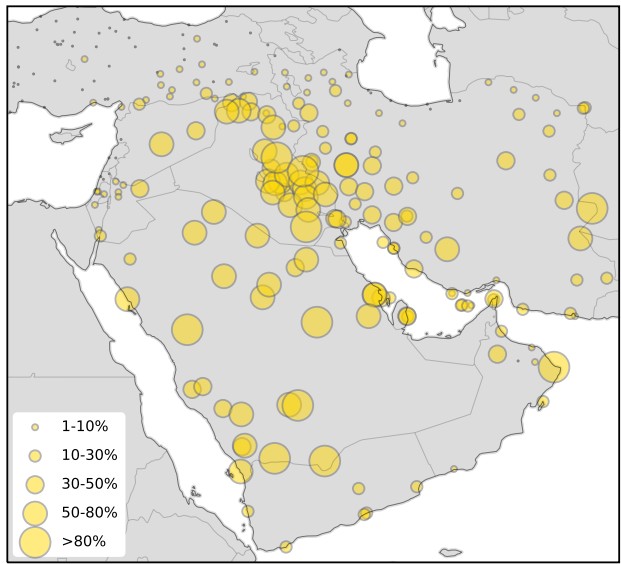

**Figure 12.** Dust event frequency ($FR$) in the Middle East based on the SYNOP present weather report during 1986–2019. Gray dots are stations with $FR < 1\%$.

speed and increased soil wetness. The subsequent positive trend in $FR$ and $VI$ is consistent with the satellite observations of increasing AOD and decreasing Angström exponent and fine mode fraction (e.g., Hsu et al., 2012; Klingmüller et al., 2016). The intensified dust activity was fueled by a prolonged dry anomaly in the Tigris-Euphrates basin, which resulted in Dust Bowl-like extreme events (Parolari et al., 2016). Specifically, Fig. 13 shows that the regional mean soil moisture has been decreasing since the turn of the century, and transitioned from above-normal to below-normal around 2006, which was described by Notaro et al. (2015) as a trigger of the dust regime shift in the region. Through multivariate analysis, Klingmüller et al. (2016) suggested that the soil moisture deficit was the primary cause of the observed dust enhancement in Saudi Arabia and Iraq from 2000 to 2015. The dominant role of low soil moisture in driving the recent dust enhancement can be further confirmed from the extremely dusty period of 2008–2012, during which the reduced wind speed was unfavorable for dust production.

Notaro et al. (2015) suggested that the recent drought in the Tigris-Euphrates basin was triggered by a climate shift to La Niña and negative Pacific Decadal Oscillation (PDO) conditions, causing widespread crop failure and vegetation loss in the Fertile Crescent and leaving the alluvial plain prone to wind erosion. Indeed, significant correlations are found between NINO 3.4 and scPDSI ($r = 0.27$, $p < 0.001$), as well as between PDO and scPDSI ($r = 0.53$, $p < 0.001$). PDO also appears to be more correlated ($r = -0.36$, $p < 0.001$) with MERRA2 DOD than NINO 3.4 ($r = -0.15$, $p < 0.005$). As seen in Fig. 13, PDO has transitioned into positive and weak negative phases since 2015, which has led to the amelioration of drought and consequently, a decrease of dust activity in the Middle East.

Comparing the role of soil moisture and wind in modulating the interannual dust variations between 1986 and 2019, it is revealed that wind speed has a higher correlation with $FR$ ($r = 0.53$, $p < 0.001$) and $VI$ ($r = 0.31$, $p < 0.001$) than does the

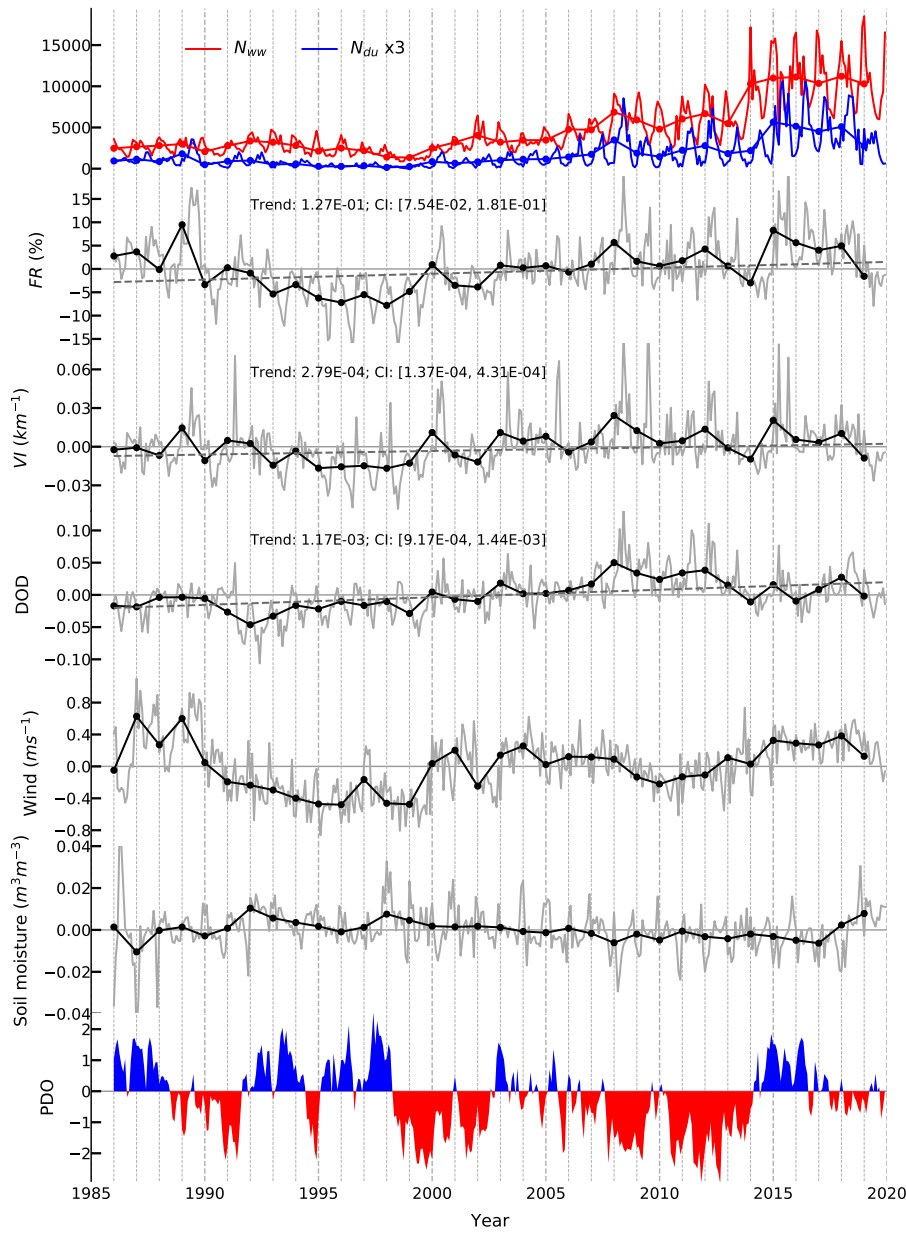

**Figure 13.** Same as Fig. 11 but for the Middle East. Dust optical depth (DOD) and soil moisture are averaged over region 13°N–38°N, 32°E–62°E. PDO stands for Pacific Decadal Oscillation.

soil moisture ($r = −0.21$ and $−0.17$, respectively). This does not contradict the above finding of soil moisture being the driving factor of the recent dust change, which is considered a component of decadal variability (as seen from the strong correlation
between PDO and scPDSI).





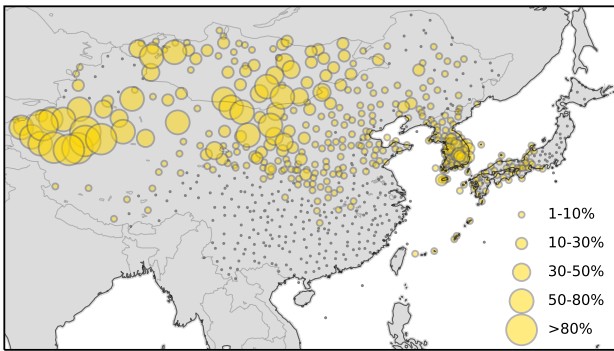

**Figure 14.** Springtime dust event frequency ($FR$) in East Asia based on the SYNOP present weather report during 1986–2019. Gray dots are stations with $FR < 1\%$.

## 5.3 East Asia

In East Asia, dust outbreak is a recurring phenomena in late winter and spring, when cyclonic cold fronts sweep across the Tarim Basin in northwest China and the Gobi occupying northern China and southern Mongolia, leading to dust storms that travel eastward to the Korean Peninsula, Japan, and sometimes across the Pacific to western United States (Qian et al., 2002).

Figure 14 shows the SYNOP weather stations included in the analysis, and depicts a west-east gradient in the springtime $FR$ extending from the Taklamakan and Gobi deserts to Japan. Under rare circumstances ($FR < 10\%$), dust can even travel south, affecting the central provinces of China.

Focusing on the spring season only, Figure 15 shows a decrease over time of $N_{ww}$ due to reduced stations and replacement of manual with automated stations (especially in Japan). Similar to North Africa, the stability of $N_{ww}$ can be improved by

screening the stations based on the record length, which however does not alter the qualitative behaviors of $FR$ and $VI$. For the period of 1986–2019, $FR$ and $VI$ show weak positive trends, but with large uncertainties, even in the sign of the trend. In contrast, the MERRA2 DOD suggests a significant positive trend. The discrepancy is most striking for the period 1986– 1997, during which opposing trends were detected between the weather stations and MERRA2. Apparently, MERRA2 failed to capture the declining trend of Asian dust during the 1980s, which has been well documented (e.g., Qian et al., 2002; Ding

et al., 2005; Hara et al., 2006; Shao et al., 2013). The dust decline was driven by the weakening of surface winds, as shown in Fig. 15 and suggested by past studies, which reported widespread decrease of surface winds in China during the 1970s– 1990s, with greatest reduction during spring and in the gusty wind segment (Guo et al., 2011; Lin et al., 2013). The wind stilling was attributed to large-scale atmospheric circulation change and/or inhomogeneous surface warming, which reduced the geopotential height difference between low- and mid-latitudes (Lin et al., 2013). Zhao et al. (2006) suggested that Asian

dust is affected by multiple teleconnection patterns which alter the surface winds, upper-level winds, and precipitation. Here, moderate correlations are found between the wind speed anomaly and Arctic Oscillation or AO ($r = -0.18$, $p < 0.01$), Niño 3.4 index ($r = 0.15$, $p < 0.01$), and PDO ($r = 0.21$, $p < 0.01$). Similar relationships, however, are not found between the climate



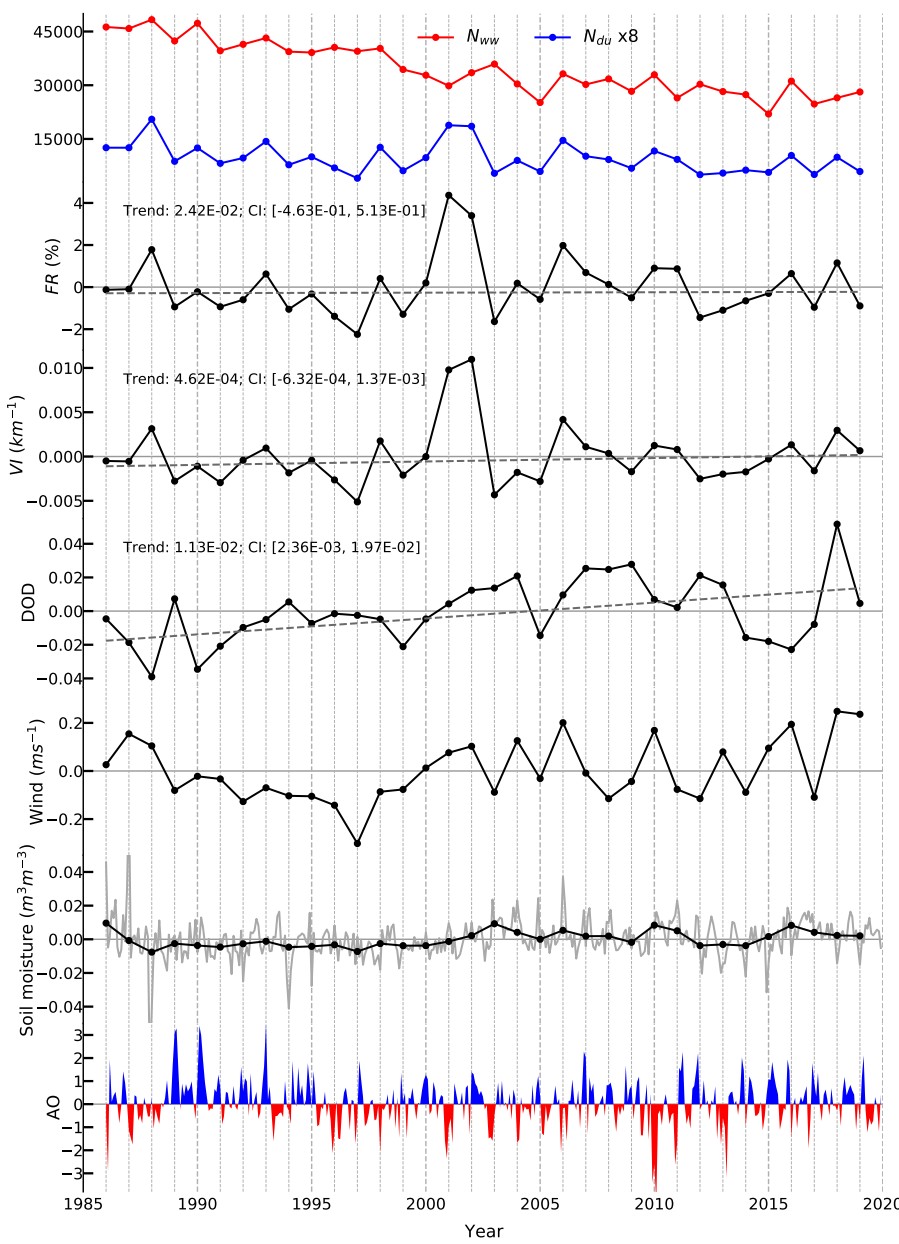

**Figure 15.** Same as Fig. 11 but for East Asia and the spring season only (except for soil moisture and AO). Dust optical depth (DOD) and soil moisture are averaged over region 36°N–48°N, 75°E–115°E. AO stands for Arctic Oscillation.

indices and soil moisture. Among the climate indices, AO has the strongest correlation with $FR$ ($r = -0.15$, $p < 0.01$) and $VI$ ($r = -0.14$, $p < 0.01$). It has been suggested that a positive winter/spring AO phase is associated with a northward shift of the





polar jet, which reduces the frequency of arctic airmass invasion into the mid-latitudes, leading to weaker East Asia winter monsoon and less dust (Gong et al., 2006; He et al., 2017).

From the above analysis and previous studies, wind appears to be most important factor in driving the variability of Asian dust (Lee and Sohn, 2009). Indeed, surface wind speed is strongly correlated with both $FR$ ($r = 0.47$, $p < 0.001$) and $VI$ ($r = 0.43$, $p < 0.001$). Given that a large number of stations in the analysis are not from dust source areas (Fig. 14), the dust-wind

correlation is expected to be even stronger near the source. As shown in Fig. 15, the springtime wind speed started to recover in 1998 after the initial decline, and have stayed relatively stable in the past two decades. In response, no significant trends are found in $FR$, $VI$, and MERRA2 DOD between 2000 and 2019. In addition to wind, vegetation has been suggested as an important player in modulating the Asian dust emission (Lee and Sohn, 2009; Jeong et al., 2011; Fan et al., 2014). Since the vegetation cover and phenology depends on precipitation and temperature, which covary with the wind pattern under the

influence of large-scale sea surface temperature or pressure anomalies, physical dust models will be needed to determine the relative importance of vegetation and wind speed in regulating the Asian dust.

## 6   Conclusions

Surface weather stations provide the longest instrumental observations of aeolian dust processes at sub-daily scale over the world's major dust sources, as well as the downstream regions impacted by transported dust. Two variables are most relevant:

present weather report from manned stations ($ww$), which contains qualitative reports of dust event occurrence, and visibility. Through an analysis of the continuity of dust record in the NOAA ISD database, it is found that the SYNOP report type provides the longest, continuous dust record since 1986 at the global scale. The SYNOP data also meet the requirement on the continuity of dust weather code usage and $ww$-visibility consistency.

Empirical analysis are performed on global and regional mean dust variations, trends, and the possible roles of climate

teleconnection patterns and physical processes. The climate teleconnection identified in this study (Fig. 11, 13, 15) highlights the most relevant large-scale processes driving the dust variations in North Africa, Middle East and East Asia. It should be noted that the teleconnection patterns may interact or covary with each other, and interact with local factors, which would require physical models to elucidate the mechanisms and feedbacks in dust-climate connections.

The main findings of this study are summarized below.

1. Due to the qualitative nature of $ww$ and the derived dust event frequency ($FR$), a new variable $VI$ is created by combining $FR$ with visibility to provide a more quantitative measure of dust burden. Compared to $FR$, $VI$ has higher correlations with the MODIS-derived dust optical depth and the dust column properties from two global aerosol reanalysis datasets (MERRA2 and CAMS).

   2. Both $FR$ and $VI$ capture the geographic distribution and relative importance of major dust sources, as well as the impact

of mega-drought events on the decadal mean dust pattern. Globally, the dust activity experienced a decline from 1986

to 1996/97, followed by a slower rebound. The global mean dust variation is correlated with the soil moisture anomaly, especially when soil moisture leads by 14 months, suggesting a lagged effect of soil wetness and/or vegetation on dust.

3. North Africa shows a declining dust trend from 1986 to 2019. However, the most recent decade is associated with a rebound of dust activity, which may be due to reduced soil moisture and increased surface winds, following the transition of the North Atlantic Oscillation (NAO) to recurring positive phases since 2011.

4. Dust in the Middle East is characterized by an initial decline from 1986 to 1998, and an increasing trend from 1998 to 2019, driven by a prolonged drought in the Tigris-Euphrates basin associated with strong negative Pacific Decadal Oscillation (PDO) modes. As PDO turned positive and weak negative after 2015, the drought became less severe which has led to a recent decrease of dust activity.

5. The spring dust activity in East Asia is primarily driven by wind. Declining surface winds were responsible for a negative dust trend from 1986 to 1997, which was suggested as a result of weakened geopotential height difference between low- and mid-latitudes. No significant dust trends were detected during the past two decades, due to relatively stable wind speed.

Overall, this study demonstrates that the surface weather station observations of dust event occurrence and visibility are an invaluable data source for characterizing large-scale dust variability on seasonal to interannual and multidecadal timescales. After a series of data screening and quality assessment of the NOAA ISD, this study has created an initial version of a global, homogenized dust-climate dataset that is suitable for wind erosion monitoring, dust source mapping, and dust-climate analysis at local to global scales. The dataset can be accessed at http://dx.doi.org/10.17632/399fd6jzm4.1.

*Data availability.*

The dust-climate dataset presented in this study will be published at http://dx.doi.org/10.17632/399fd6jzm4.1.

*Author contributions.* X. Xi is the sole contributor to this study and manuscript.

*Competing interests.* The author declares no conflict of interest

*Acknowledgements.* The author acknowledges William Brown (NOAA NCEI) for assisting with the use of NOAA ISD. The ISD data are obtained from https://www.ncei.noaa.gov/data/global-hourly/. Other publicly available data used in this study include the dust aerosol optical depth from MODIS (https://doi.pangaea.de/10.1594/PANGAEA.909140); CAMS global reanalysis (https://atmosphere.copernicus.eu/data);



MERRA2 global reanalysis (https://gmao.gsfc.nasa.gov/reanalysis/MERRA-2/); drought index (https://crudata.uea.ac.uk/cru/data/drought/); and climate index data (https://psl.noaa.gov/data/climateindices/).



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
