# Peer review of "Global aeolian dust variations and trends: a revisit of dust event and visibility observations from surface weather stations"

_Atmospheric Chemistry and Physics, 2020_

## Short Comment (SC1) · 13 Oct 2020

Thanks very much for this interesting work. I welcome the revisitation of the global dust studies as satellite and modeling improve, it is always needed to reevaluate the subject. Also, big picture analysis of dust activity in the context of climatic indexes as shown here are always welcomed.

Without providing a long review, I'd like to add a few comments and clarifications which I believe are in order and if the author (or editor) thinks so, I think it would strengthen the paper.

My main comment is that the title of the paper conveys an idea that it is not quite consistent with the evidence shown in the paper. Largely missing in this paper is the subject of dust activity at high latitudes (HLD for short) and with this regard, this is an important omission of an important development in the last few years with regards to global dust characterization. Here I list a few sources of information regarding HLD that illustrate my points:

A.1 A major review on the subject was published by Bullard et all (2016). In addition, a large database of publications on the subject can be found here https://icedustblog.wordpress.com/publications/ (mostly references related to Greenland and Iceland dust activity)

A.2 The satellite data used in this study (MODIS data from the Voss and Amato database) is only characteristic of observations from 45S to 50N , thus it does not include any of the high latitude dust sources reported in the studies from the previous section.

A.3 Surface Visibility and Synop codes have been successfully used to characterize high latitude dust activity in Iceland (see above blog for references) and in Patagonia (Gasso and Torres, 2019, Gasso et al, 2010, Gasso and Stein, 2007)

I believe that just an appropriate adjustment of the title is needed in order to reflect that this study is not global.

In addition, where are a few clarifications that jumped out when reading the paper:

B.1) What is the time resolution in the model-satellite comparison? specifically is the model sampled at the same time of the model overpass? This information would be useful to guide future research based on your analysis.

B.2) What is the density of stations used in this study? what regions are not well captured by the surface and satellite data?

B.3) In figure 4, there is a singular dot in South America, possibly in Chile or Argentina.

From my own work, I am familiar with the station Tinogasta in Argentina (by the Andes mountains where this dot is located). this is station is consistently biased to report more dust activity than actually is. I found this out by talking to Argentina's weather bureau central data archive manager. I suggest removing such point.

B.4) Is there a consideration for the fact that a large amount of dust activity occurs in cloudy conditions? Satellite polar observations are biased low not only because 1-2 obs per day in a given cite but also because cloudiness, which tends to be more pervasive towards higher latitudes. See Gassó and Torres (2019) for more on this.

I believe that all these issues can be easily addressed and turn this fine work into a more complete study.

Santiago Gassó, GESTAR/NASA santiago.gasso@nasa.gov
http://science.gsfc.nasa.gov/sed/bio/santiago.gasso

References

Bullard J.E., Baddock, M., Bradwell, T., Crusius, J., Darlington, E., Gaiero, D., Gassó, S., Gisladottir, G., Hodgkins, R., McCulloch, R., McKenna Neuman, Ch., Mockford, T., Stewart, H., Thorsteinsson, Th., 2016. High Latitude Dust in the Earth System. Reviews of Geophysics: DOI: 10.1002/2016RG000518.

Gassó, S., & Torres, O. (2019). Temporal characterization of dust activity in the Central Patagonia desert (years 1964–2017). Journal of Geophysical Research: Atmospheres, 124, 3417– 3434. https://doi.org/10.1029/2018JD030209

Voss, K. K., and A. T. Evan, 2020: A New Satellite-Based Global Climatology of Dust Aerosol Optical Depth. J. Appl. Meteor. Climatol., 59, 83–102, https://doi.org/10.1175/JAMC-D-19-0194.1.

Gassó, S., and A. F. Stein. 2007. "Does dust from Patagonia reach the sub-Antarctic Atlantic Ocean?" Geophysical Research Letters 34 (1): L01801 [10.1029/2006GL027693]

Gassó, S., A. Stein, F. Marino, et al. 2010. "A combined observational and modeling approach to study modern dust transport from the Patagonia desert to East Antarctica." Atmos. Chem. Phys. 10 (17): 8287-8303 [10.5194/acp-10-8287-2010]

---

## Referee Comment (RC1) · Anonymous Referee #2 · 8 Nov 2020

**Review report acp-2020-813**

The current study deals with the analysis of dust long-term variability over the major deserts of the planet and the main downwind regions. To realize, observations from weather stations, derived by the NOAA Integrated Surface Database (ISD), have been analyzed. Overall, it is a very nice and comprehensive study which fits very well to the issues covered by the Atmospheric Chemistry and Physics journal. Moreover, the structure of the work is well defined and the manuscript it is very well written. Therefore, I recommend the submitted study to be published after taking into account the following minor comments and suggestions.

1. **Lines 50-55:** Here you have to add satellite sensors (SEVIRI, CATS, IASI, CALIPSO) which have been widely used in various studies focusing on dust aerosols.
2. **Lines 63-65:** It is missing a discussion about the advantages of passive satellite sensors (e.g. spatial coverage) as well as few sentences about the benefits and the drawbacks of active satellite sensors (e.g. CALIOP).
3. **Line 121:** Could you please explain better this sentence?
4. **Equation 1:** Please provide a definition for the FR abbreviation.
5. **Lines 167 – 173:** MERRA-2 winds have been also used for the sea-salt aerosols.
6. **Lines 177 – 180:** Few corrections must be implemented here. MODIS radiances, instead of AOD, are used and are transformed to bias-corrected AOD with respect to AERONET. MISR AODs, without bias correction, are assimilated in MERRA-2 only over bright surfaces whereas AERONET measurements are taken into account until 2014.
7. **Line 187:** What do you mean "*even false recording*"? If it is not reliable, then why it is considered in the analysis?
8. **Figure 3:** It is hard to distinguish among the curves and the dashed lines. It is better to produce separate plots (for individual codes or groups) and give the all-time averages in a table or insert them in the plots.
9. **Figure 7:** Which MODIS DOD are you using? From Voss and Evan (2020)? Have you tried to reproduce the same plot but considering only the coincident measurements among MODIS, CAMS and MERRA-2 for the common period? Please keep in mind that MODIS provides single measurements per day whereas the reanalyses datasets take into account the diurnal cycle. Can you comment on this? Moreover, it is needed an explanation of how the global means have been calculated. In Figure 5 in Levy et al. (2009), it is evident that the calculation of the domain averages is affected by the selected approach. This is quite critical for the satellite data in which there are gaps due to the inability of the applied algorithm to provide a retrieval.
10. **Figure 8:** Please consider to reproduce the plot with the collocated data for the common period.
11. **Tables 3 and 4:** It would be interesting to include also other dust optical depth databases such as MIDAS (Gkikas et al., 2020) and LIVAS (Amiridis et al., 2015).
12. **Lines 305 – 306:** As I have already mentioned above, MERRA-2 assimilates MISR AODs above bright surfaces. Likewise, in MERRA-2 the anthropogenic dust sources are not considered.
13. **Figure 9:** I would like to see the results at station level. More specifically, three global maps are needed with the stations colored based on the correlation coefficient of VI with scPDSI, soil moisture and wind.
14. **Lines 324-325:** The vertical structure of the dust layers plays a key role when attempting to compare spaceborne retrievals with near-surface observations (see Section 4.4 in Gkikas et al. (2016)).
15. **Line 338:** How the global means from the weather stations are calculated? Are you using any weight based on the data availability?
16. **Figures 11, 13 and 15:** Same comment as for Figure 9. In addition, please use always red and blue color for the positive and negative phases, respectively, for the teleconnection patterns.

---

## Referee Comment (RC2) · Anonymous Referee #3 · 12 Nov 2020

The paper is potentially publishable, but their are many issues in both the approach and the presentation that need to be improved before it can be published.

1. The author is not careful about describing the quality of the data analyzed in the paper in the methods section. All 'observational' and model data should include some information about the quality from the literature. A. All of these records are very qualitative, of course. Are there any studies which show that visibility can be used as a proxy? please cite and examine how much you can conclude from qualitative data, or don't use the station data. B. MERRA model output is presented as if it were observations. "CAMS and MERRA2 represent recent advances in developing atmospheric

composition reanalysis using global model systems with capabilities to assimilate satellite observations of atmospheric aerosols and gaseous species (Gelaro et al., 2017; Inness et al., 2019)." The way you describe this, it is as if you think these are assimilations of aerosols, but really very little aerosol data is assimilated and they tend not to do very well against the observations, so please discuss the discrepancies and use these model results NOT as observations but as model results. For example: https://acp.copernicus.org/articles/20/10047/2020/.

2. I also do not think there is value in presenting all the codes in the first few graphs. Do they really have different correlations or trends? If you only showed 2 of these codes, would we learn less? Please think carefully what your take-away messages are and if we would actually learn more if you presented less. 3. Pretty much all the tables and figures have NO description of what is in the table and figures in the figure captions. Please describe what is actually plotted clearly. I probably misunderstood most of the plots and would need to re-review once the plots are explained. 4. The author wants to indicate that drought caused some of the dust, but doesn't show statistical studies of this relationship. Please add in any mechanisms that you want to evaluate and make sure your results are statistically significant.

"Figure 1. Analysis of the continuity of present weather (ww) reports in ISD. (a) Global monthly number of stations (Nstn), number of ww reports (Nww), and number of dust event reports (Ndu). For clarity Ndu is multiplied by 30; (b) Nww from different report types."

"Therefore, the SYNOP data between 1986 and 2019 are most suitable for global-scale dust analysis." I assume this means you will only present the SYNOP data in this paper. This figure can go into the supplement, and say this in the methods instead. But wait, you are using the other times periods, after you say that you shouldn't use them? Seems odd??? Please justify their use, and then discuss how that changes your results whenever you use the data that isn't very consistent.

Table 2 is really unreadable because we can't understand what these codes are. I think in Table 1 you should come up with some short acronym that describes each of these codes, and use it for the rest of the paper, so that we know which code is which and what they mean. Or just exclude this whole section, as not really very important or interesting compared to the others.

"Figure 2. Analysis of the continuity of dust weather code usage. From top to bottom: monthly number of reports of all and individual dust codes in the global SYNOP data. Horizontal lines are all-time averages. Dust weather codes are described in Table 1". This figure caption does not tell us what is plotted. Please describe the variable plotted. Is this the number of observations? Why do you present so many? Are they actually different? The same? If they are the same, then show one. If they are different, tell us how they are different and why.

"Figure 3. Temporal consistency of the harmonic mean visibility associated with dust weather codes. Dash lines are all-time averages. Dust weather codes are described in Table 1." What variable is presented in figure3? It should be described in the figure caption. Is this the number of observations? Or the correlation? Do you really need to present all the variables, or could you just show one and the offset between them?

"Figure 3 shows that while there are significant year-to-year fluctuations in the harmonic mean visibility associated with dust codes, they generally fall into three clusters: ww = 06–09 (3.7 km), ww = 30–32, 98 (1.5 km), and ww = 33–35 (0.7 km)." please tell use why this is important. Does this make sense? Etc. right now there is no context for this statement.

"Figure 4. Decadal mean dust event frequency (FR). Gray dots are stations with FR < 1%." Please indicate in the figure caption exactly which data you use to define this, since there are many ways to do this. Repeat information in the methods to be clear. Your figure captions are so brief as to make reading your paper much more difficult than it needs to be.

"Using Eq. 1, the decadal mean FR 260 is calculated for stations with at least 5 years' data in each decade, as shown in Fig. 4." Please don't write sentences like this that are difficult to understand. Tell us again what equation 1 uses to calculate the frequency of occurrence in English. (for example: Using dust frequency as calcualted from the visibilty data (equation 1), we evaluate the decadal values...)

These two paragraphs are very hand wavey, but on important, and easily plottable points: "The distinctive changes of decadal mean FR can be linked to multidecadal climate variations, especially the occurrence of mega-drought events lasting several years or even decades. Elevated dust activity can be observed in areas affected by persistent drought, where the reduction of soil moisture and vegetation leaves the exposed, dry soil prone to wind erosion. For example, a striking feature in the 1950s is the widespread, frequent dust events in the U.S. Southwest and Midwest, with several stations reporting FR > 20% in the High Plains of Texas and Colorado. The heightened dust activity was fueled by a 11-year-long (1946–1956) drought that afflicted a massive area centered in the Southwest U.S. (Fye et al., 2003). The 1950s drought was characterized by a prolonged lack of precipitation and excessive warm temperatures, which caused crop failure and livestock feed shortage (Goudie and Middleton, 1992). As the drought came to an end in the spring of 1957, FR started to decline and has since remained low in the last 50 years. Similarly, North Africa experienced progressively drier conditions during the 1970–80s in the Sahel, a semiarid dryland belt at the southern border of Sahara Desert (Giannini et al., 2008). The Sahelian drought was triggered by anomalous sea surface temperature (SST) in the tropic Atlantic and Indian Ocean (Dai, 2011). The Sahelian dust frequency during drier-than-normal years, especially in the 1980s when drought was most severe, is significantly higher compared to the pre- or post-drought periods. The drought-induced dust enhancement is also evident from the frequent dust weather observed downstream, including 285 the Caribbean, Gulf of Mexico, and Iberian Peninsula. This is consistent with the long-term in situ dust measurements in Barbados and Miami, Florida, indicating a positive correlation between the Sahel dry anomaly and African dust outflow across the tropical

North Atlantic (Prospero and Lamb, 2003; Zuidema et al., 2019). With the amelioration of Sahelian drought in the 2000s, FR experienced significant decreases at the source and downwind, consistent with ground and satellite observations (Hsu et al., 2012; Li et al., 2014). In the past decade, increased dust activity can be observed in West Africa and the Middle 290 East, which will be discussed later." Please show this is true statistically significantly, in a clear way. perhaps show the 1-sigma and 2-sigma bounds and highlight the time periods above or below, or do a correlatoin.

"if weather stations provide a consistent view of global dust variations, FR and VI are compared with the datasets described in Sect. 2.3." You can compare the dust variations to satellite data and AERONET data and evaluate the dust variables, but please do not pretend that MERRA output is more than model output.

"Figure 6. Global monthly (gray) and annual (black dotted) FR and VI . Horizontal lines are all-time averages." Please describe what you are plotting in complete detail, including repeating which variables are used, etc, so that your figures are self-standing. How are you averaging over the globe when you only have spotty data? Needs to be described in the methods section, and make sure you are doing this in a manner that is consistent with the observations and models you are comparing again (for example, pick each point from the dust station data, and match to the model output at the same grid box, so are weighting similarly).

"Trends are further calculated from the monthly anomalies using the pyMannKendall package developed by Hussain and Mahmud (2019), which consists of multiple Mann-Kendall test options to accommodate the seasonality and serial correlation in the data. The Mann-Kendall test is a non-parametric test of the presence of monotonic trend in the data, and has advantage 330 over parametric methods (e.g., t test) for its insensitivity to outliers, missing values, and the statistical distribution of the data. The Mann-Kendall test is designed for serially independent data and thus can be influenced by the presence of autocorrelation in the data, which either increases the uncertainty of estimated trends or prolongs the length of time period required to detect a given trend

(Weatherhead et al., 1998)." All methods should be in the methods section, or in figure captions, not in the results section.

"The decadal mean FR and VI (Fig. 4 and 5) indicate mega-drought events are associated with extremely active dust periods in the 20th century." To make this statement you need to have compared against precipitation or P-E data and show a statistically significant change/relationship

"Dust variation in North Africa from 1986 to 2019. From top to bottom: monthly Nww and Ndu, and anomalies of FR, VI , MERRA2 DOD, wind speed, soil moisture, and the Jones North Atlantic Oscillation (NAO) index. DOD and soil moisture are averaged over region 10°N–20°N, 20°W–20°E; Dotted curves are annual averages. Dashed lines are Theil-Sen linear regression of FR, VI and DOD, with the annual trends and confidence intervals (CI) shown." What is Nww? Ndu? Should be explained in the figure caption, as well as FR, VI.

"Consistent with previous studies, Fig. 11 shows that African dust was at a historic high during the late 1980s due to abnormally strong winds and low soil moisture, both favorable for dust production (e.g., Prospero and Lamb, 2003; Chiapello et al., 2005; Evan et al., 2016)." Is this statistically significantly true? Please check or put your statistical significance on the plot.

"Indeed, significant correlations are found between NINO 3.4 and scPDSI (r = 0.27, p < 0.001), as well as between PDO and scPDSI (r = 0.53, p < 0.001). PDO also appears to be more correlated (r = −0.36, p < 0.001) with MERRA2 DOD than NINO 3.4 (r = −0.15, p < 0.005). As seen in Fig. 13, PDO has transitioned into positive and weak negative phases since 2015, which has led to the amelioration of drought and consequently, 435 a decrease of dust activity in the Middle East." Is this based on the time series in the plots? Please specify.

"The dust decline was driven by the weakening of surface winds, as shown in Fig. 15 and suggested by past studies, which reported widespread decrease of surface winds

in China during the 1970s– 1990s, with greatest reduction during spring and in the gusty wind segment (Guo et al., 2011; Lin et al., 2013)." Again, do not show your results without showing that they are statistically significant. Please show a correlation coefficient and that it is significant before you make such important, but unclear statements.

---

## Author Comment (AC1) · 12 Nov 2020

*Thank you very much for the constructive comments. Please see my replies below.*

**Review report acp-2020-813**

The current study deals with the analysis of dust long-term variability over the major deserts of the planet and the main downwind regions. To realize, observations from weather stations, derived by the NOAA Integrated Surface Database (ISD), have been analyzed. Overall, it is a very nice and comprehensive study which fits very well to the issues covered by the Atmospheric Chemistry and Physics journal. Moreover, the structure of the work is well defined and the manuscript it is very well written. Therefore, I recommend the submitted study to be published after taking into account the following minor comments and suggestions.

1. **Lines 50-55:** Here you have to add satellite sensors (SEVIRI, CATS, IASI, CALIPSO) which have been widely used in various studies focusing on dust aerosols.

   *Response: this section is not intended to give a comprehensive review of satellite aerosol sensors, but briefly mentions a few with long-term observations of dust burden needed for the analysis of interannual to multidecadal dust variations (the objective of this study). Passive sensors with a wide swath and sufficiently long records are considered more suitable for such purpose. Among those, only a few provide aerosol information over deserts.*

   *It now reads "For contemporary dust, satellite remote sensing has greatly advanced the monitoring capability of large-scale dust events on the daily basis. Specifically, the long-term aerosol climatology derived from low- and moderate-resolution passive sensors in sun-synchronous orbits remain the most widely used datasets to track long-term dust trends and variations at global and regional scales. To name a few, the Advanced Very High Resolution Radiometer (AVHRR) sensors onboard NOAA's weather satellite series provide the longest over-ocean aerosol optical depth (AOD) dataset since the 1970s, available under the NOAA Climate Data Record program (Zhao et al., 2008). As a semi-quantitative measure of column aerosol burden, the absorbing aerosol index (AAI) derived from the UV bands of Total Ozone Mapping Spectrometer (TOMS) and similar instruments allows dust detection over desert surfaces, and the derivation of a global dust source map for improving dust model simulations (Ginoux et al., 2001; Prospero et al., 2002). Under the NASA Earth Observing System (EOS) program, newer instruments designed specifically for measuring the atmospheric composition with improved sensor characteristics, notably the twin Moderate Resolution Imaging Spectroradiometer (MODIS) sensors aboard Terra and Aqua, have further expanded the capability of large-scale dust characterization and source mapping from space (Ginoux et al., 2012). Many other instruments, including those from European satellites which are being reprocessed under the ESA Climate Change Initiative, offer a complementary view of the dust aerosol burden and properties since the 1990s, mostly over downwind areas where elevated dust layers can be effectively distinguished from the dark background of vegetation or ocean surfaces (Popp et al., 2019)".*

2. **Lines 63-65:** It is missing a discussion about the advantages of passive satellite sensors (e.g. spatial coverage) as well as few sentences about the benefits and the drawbacks of active satellite sensors (e.g. CALIOP).
   *Response: see my reply to #1 above. It now reads "While spaceborne sensors offer an unprecedented view of large-scale dust events, they can be limited by the difficulty of separating dust from the total aerosol signal, limited sampling frequency in sun-synchronous orbits (usually once per day), incapability of detecting dust under clouds, and sensor calibration instability. When it comes to the analysis of interannual to multidecadal dust variations governed by low frequency climate variability, satellite records are further limited by the relatively short length*

*and cross-sensor consistency and continuity, especially over dust source areas where quantitative information of the column aerosol burden is available only for the last two decades."*

3. **Line 121:** Could you please explain better this sentence?
*Response: In the manual report of present weather (ww), the code varies from 00 (lowest priority) to 99 (highest priority). During the data preprocessing, I follow this rule by keeping the highest value of the ww records in ISD; However, if there is a dust code in the ww values, the dust code is kept, instead. ISD has 7 place holders for ww reports at any given time, but for most of the time, only has 1 ww report, so the above rule does not apply. When multiple ww reports do occur, the above rule makes sure that the dust event is counted and not overwritten by higher-priority events. It is rare, however, that a dust event is reported at the same time with other higher-priority weather events, specifically fog and precipitation.*

4. **Equation 1:** Please provide a definition for the FR abbreviation.
*Response: FR is the dust event frequency, expressed in the percentage of dust event reports out of all ww reports. In the revised manuscript, FR will be abbreviated as "f", while VI will be replaced by the dust extinction coefficient, $\beta$, which is more intuitive and easier to interpret.*

5. **Lines 167 – 173:** MERRA-2 winds have been also used for the sea-salt aerosols.
*Response: It now reads "To separate the dust component over ocean, Voss and Evan (2020) followed Kaufman et al. (2005) by removing the contribution of fine-mode (anthropogenic and biomass burning) and sea salt aerosols (derived from reanalyzed winds) from the total AOD."*

6. **Lines 177 – 180:** Few corrections must be implemented here. MODIS radiances, instead of AOD, are used and are transformed to bias-corrected AOD with respect to AERONET. MISR AODs, without bias correction, are assimilated in MERRA-2 only over bright surfaces whereas AERONET measurements are taken into account until 2014.
*Response: Thanks for pointing it out. It now reads "MERRA2 is produced by the Goddard Earth Observing System version 5 (GEOS-5) which simulates dust emission and dispersion using the Goddard Chemistry, Aerosol, Radiation, and Transport model (GOCART), and assimilates multiple aerosol products, including bias-corrected AOD derived from the observed radiances by AVHRR (over ocean only, until 2002) and MODIS (over ocean and dark surfaces, from Terra since 2000 and Aqua since 2002), MISR AOD over bright land surfaces (surface albedo > 0.15, without bias correction), as well as ground-based AOD measurements from AERONET until 2014 (see details in Randles et al., 2017)."*

7. **Line 187:** What do you mean "*even false recording*"? If it is not reliable, then why it is considered in the analysis?
*Response: Removed.*

8. **Figure 3:** It is hard to distinguish among the curves and the dashed lines. It is better to produce separate plots (for individual codes or groups) and give the all-time averages in a table or insert them in the plots.
*Response: Fig. 3 has been remade. See below.*

[Figure]

9. **Figure 7:** Which MODIS DOD are you using? From Voss and Evan (2020)? Have you tried to reproduce the same plot but considering only the coincident measurements among MODIS, CAMS and MERRA-2 for the common period? Please keep in mind that MODIS provides single measurements per day whereas the reanalyses datasets take into account the diurnal cycle. Can you comment on this? Moreover, it is needed an explanation of how the global means have been calculated. In Figure 5 in Levy et al. (2009), it is evident that the calculation of the domain averages is affected by the selected approach. This is quite critical for the satellite data in which there are gaps due to the inability of the applied algorithm to provide a retrieval.

*Response: The MODIS dust optical depth (DOD) in Fig.7 is from Voss and Evan (2020). The figure caption is revised to reflect that. The global DOD mean is calculated without taking into account the data match in time or space. Collocation would be needed for a critical comparison of MODIS vs. reanalysis, which is outside the scope of this study. Rather, the DOD mean is calculated simply as the domain average of the monthly data of Voss and Evan (2020), CAMS, and MERRA2, as one would normally do when using them for global analysis. The manuscript does include discussions on the disparity of MODIS and model reanalyzed AODs: "The global average DOD is calculated from the MODIS DOD (550 nm) by Voss and Evan (2020) and the dust extinction optical depth (550 nm) from the CAMS and MERRA2 reanalyses, without taking into account the collocation in space and time. Figure 7 reveals systematic differences between the DOD datasets. For the overlapping period (2003–2018), the mean DOD is 0.042, 0.013 and 0.024 from MODIS, CAMS and MERRA2, respectively. CAMS also appears to be less variable than MODIS and MERRA2. The offsets in the mean DOD can be attributed to a number of factors. For example, MODIS DOD is based on once-per-day observations over both ocean and land, including deserts where large AOD values (up to 5) are allowed in the Deep Blue algorithm, whereas CAMS and MERRA2 are generated by global models which are capable of simulating the diurnal cycle of dust loading and assimilating satellite aerosol observations, including over deserts. Specifically, CAMS assimilates the MODIS Deep Blue AOD without bias correction (Inness et al., 2019). Unlike CAMS, MERRA2 does not use the Deep Blue products, but rather assimilates the MISR AOD without bias correction over desert surfaces (Randles et al., 2017). In addition, the models used to produce CAMS (ECMWF IFS) and MERRA2 (NASA GEOS-5) have a variety of differences in the model configuration, dust parameterization, dust optical properties, meteorological input, and AOD assimilation, all of which may contribute to the discrepancy between CAMS and MERRA2 DOD."*

10. **Figure 8:** Please consider to reproduce the plot with the collocated data for the common period. *Response: See my response to #9. Fig. 8 is to compare the multi-year average annual dust cycle over the common period of 2003-2018, based on all available information from each*

*dataset.*

11. **Tables 3 and 4:** It would be interesting to include also other dust optical depth databases such as MIDAS (Gkikas et al., 2020) and LIVAS (Amiridis et al., 2015).

    *Response: Thanks for the suggestion. I will consider including these datasets in an upcoming manuscript as an extension of this study.*

12. **Lines 305 – 306:** As I have already mentioned above, MERRA-2 assimilates MISR AODs above bright surfaces. Likewise, in MERRA-2 the anthropogenic dust sources are not considered.

    *Response: Fixed. See my response to #9.*

13. **Figure 9:** I would like to see the results at station level. More specifically, three global maps are needed with the stations colored based on the correlation coefficient of VI with scPDSI, soil moisture and wind.

    *Response: I agree that station level analysis would provide information on the dust-climate relationship at the local scale. However, I think using global aggregated data is sufficient for studying the global mean behavior of dust in response to climate, as shown in the correlation reanalysis. The same is also true for the regional level analysis for North Africa, Middle East and East Asia. In addition, the dust-climate connection may not be properly manifested at station levels, because most stations are impacted by transported events from upwind sources and therefore, the climate variables (wind, soil wetness) observed at those stations likely do not properly represent the climate conditions driving the dust variations. On the other hand, the stations located near dust source areas are complemented by downwind observations to create a full long-term dust record, given that those stations have a great chance of interruption and discontinuity.*

14. **Lines 324-325:** The vertical structure of the dust layers plays a key role when attempting to compare spaceborne retrievals with near-surface observations (see Section 4.4 in Gkikas et al. (2016)).

    *Response: Thanks for pointing this out. It now reads "weather stations focus on dust conditions in the lower boundary layer, whereas satellites retrieve the total aerosol column and are unable to detect aerosols below clouds. As a result, the variable vertical profile of dust can affect the relationship between the ambient dust condition (f and β) and total column dust burden.".*

15. **Line 338:** How the global means from the weather stations are calculated? Are you using any weight based on the data availability?

    *Response: the global mean from weather stations is calculated by aggregating the weather reports from all eligible stations on the monthly basis (see Eq. 1). No weighting is applied.*

16. **Figures 11, 13 and 15:** Same comment as for Figure 9. In addition, please use always red and blue color for the positive and negative phases, respectively, for the teleconnection patterns.

    *Response: see my response to #9. For the color scheme of climate indices, blue represents wet conditions, and red represents dry conditions in the study region. I can make the change as suggested.*

---

## Author Comment (AC2) · 12 Nov 2020

*I thank Dr. Santiago Gassó for his kind words and comments. My responses are given below.*

Thanks very much for this interesting work. I welcome the revisitation of the global dust studies as satellite and modeling improve, it is always needed to reevaluate the subject. Also, big picture analysis of dust activity in the context of climatic indexes as shown here are always welcomed. Without providing a long review, I'd like to add a few comments and clarifications which I believe are in order and if the author (or editor) thinks so, I think it would strengthen the paper. My main comment is that the title of the paper conveys an idea that it is not quite consistent with the evidence shown in the paper. Largely missing in this paper is the subject of dust activity at high latitudes (HLD for short) and with this regard, this is an important omission of an important development in the last few years with regards to global dust characterization. Here I list a few sources of information regarding HLD that illustrate my points:

A.1 A major review on the subject was published by Bullard et all (2016). In addition, a large database of publications on the subject can be found here https://icedustblog.wordpress.com/publications/ (mostly references related to Green-land and Iceland dust activity)

*Response: HLD is an important topic that deserves a special look, due to the unique geomorphology of high-latitude sources and local impact of HLD, as demonstrated by Dr. Gassó and his colleagues. Due to the length limit of the manuscript, I only include three most important dust source areas (North Africa, Middle East, East Asia) in section 5. Many other areas, including Central Asia, North, Central, and Southern America, and Australia, are not included. But, the global analysis in section 4 is based on the aggregated data from the entire world, including the stations of high latitudes which meet the quality control requirement described in the paper. See Fig. 4 and 5 for the decadal mean dustiness at the station level, which include a number of stations in high latitudes (e.g., Iceland). The dust dataset generated from this study has been made publicly available at [https://github.com/ixnix/duISD](https://github.com/ixnix/duISD), which includes all stations around the world, include high latitudes. Anyone is welcome to use the dataset to look at their region of interest.*

A.2 The satellite data used in this study (MODIS data from the Voss and Amato database) is only characteristic of observations from 45S to 50N, thus it does not include any of the high latitude dust sources reported in the studies from the previous section.

*Response: The MODIS dust optical depth (DOD) dataset by Voss and Amato (2020) only covers low- and mid-latitude areas, which have the most important dust sources in terms of the contribution to annual mean dust emission. As shown in Table 3, the MODIS DOD has a fairly strong correlation with the global mean dust event frequency and extinction coefficient derived from weather stations.*

A.3 Surface Visibility and Synop codes have been successfully used to characterize high latitude dust activity in Iceland (see above blog for references) and in Patagonia (Gasso and Torres, 2019, Gasso et al, 2010, Gasso and Stein, 2007)
I believe that just an appropriate adjustment of the title is needed in order to reflect that this study is not global.

*Response: see my response to A.1.*

In addition, where are a few clarifications that jumped out when reading the paper:

B.1) What is the time resolution in the model-satellite comparison? specifically is the model sampled at the same time of the model overpass? This information would be useful to guide future research based on your analysis.B.2) What is the density of stations used in this study? what regions are not well captured by the surface and satellite data?

*Response: The comparison between MODIS-derived and model reanalyzed DOD does not take into account the collocation in space and time. They global mean DOD shown in Fig. 7 is calculated simply as the global average of each dataset, as one normally would do in using these datasets in global analysis. A collocation-based comparison of MODIS vs. models is most useful to evaluate the performance of model reanalysis, which is outside the scope of this study. The station map is given in Fig S1 and S2, and can be also seen in Fig. 4 and 5. Comparing Fig. 10 of Voss and Amato (2020) with Fig. 4 of this study shows that weather stations and satellite both have good coverage of North Africa, Middle East, East Asia, Central Asia, and others, which explains the good agreement between them in the interannual variations (Table 3).*

B.3) In figure 4, there is a singular dot in South America, possibly in Chile or Argentina. From my own work, I am familiar with the station Tinogasta in Argentina (by the Andes mountains where this dot is located). this is station is consistently biased to report more dust activity than actually is. I found this out by talking to Argentina's weather bureau central data archive manager. I suggest removing such point.

*Response: Thanks for pointing this out. After a quick check, station Tinogasta (AR) is not included in Fig. 4 and 5.*

B.4) Is there a consideration for the fact that a large amount of dust activity occurs in cloudy conditions? Satellite polar observations are biased low not only because1-2 obs per day in a given cite but also because cloudiness, which tends to be more pervasive towards higher latitudes. See Gassó and Torres (2019) for more on this. I believe that all these issues can be easily addressed and turn this fine work into a more complete study.

*Response: There is no doubt that clouds is one factor among many that contribute to the disagreement between satellite-derived dust record and surface weather stations, as well as models. In section 4.2 the manuscript includes some discussions on the comparison. The global mean dustiess variation resembles that of North Africa as the world's biggest dust source, where satellites are less affected by clouds than in mid- and high-latitudes.*

---

## Author Comment (AC3) · 13 Nov 2020

*I thank the reviewer for taking the time to review this manuscript. I will try to clarify some of the confusions revealed by this review. Please see my responses below.*

1. The paper is potentially publishable, but their are many issues in both the approach and the presentation that need to be improved before it can be published.1. The author is not careful about describing the quality of the data analyzed in the paper in the methods section. All 'observational' and model data should include some information about the quality from the literature.
A. All of these records are very qualitative, of course. Are there any studies which show that visibility can be used as a proxy? please cite and examine how much you can conclude from qualitative data, or don't use the station data.

*Response: The manuscript has lengthy descriptions (Section 2) of the data source used, including present weather and visibility from weather stations, as well as three external dust optical depth datasets from MODIS (by Voss and Amato, 2020) and CAMS and MERRA2 reanalyses. Evaluation of the station data quality is fully described in Section 3. It has been mentioned in a number of places in the manuscript that weather stations have long been used for dust monitoring (e.g., two previous global studies by Shao et al. 2013 and Mahowald et al., 2007), and that the visibility data, when constrained by concurrent dust event reports, have been previously used to estimate ambient dust concentrations around the world (see Section 1 last paragraph and Section 2.1, 2.2).*

B. MERRA model output is presented as if it were observations. "CAMS and MERRA2 represent recent advances in developing atmospheric composition reanalysis using global model systems with capabilities to assimilate satellite observations of atmospheric aerosols and gaseous species (Gelaro et al., 2017; Inness et al., 2019)." The way you describe this, it is as if you think these are assimilations of aerosols, but really very little aerosol data is assimilated and they tend not to do very well against the observations, so please discuss the discrepancies and use these model results NOT as observations but as model results. For example:
https://acp.copernicus.org/articles/20/10047/2020/.

*Response: Section 2.3 clearly describes that CAMS and MERRA2 are aerosol reanalysis generated from global models, which have the capability of assimilating satellite observations from AVHRR, MODIS, MISR, AERONET, and AATSR, with or without bias corrections. The sheer volume of these data is anything but "very little". Using observation-constrained model simulations, rather than free runs, is expected to capture some of the observed dust variability and fill the data gaps caused by limited satellite overpass and clouds, etc. The discrepancy between the surface stations, satellite (MODIS), and model reanalysis is discussed in Section 4.2, Table 3 & 4, and Fig. 7 & 8.*

2. I also do not think there is value in presenting all the codes in the first few graphs. Do they really have different correlations or trends? If you only showed 2 of these codes, would we learn less? Please think carefully what your take-away messages are and if we would actually learn more if you presented less.

*Response: The qualitative dust weather reports (ww=06-09, 30-35, 98) are used to (1) derive the dust event frequency (FR, in percentage), and (2) combine with visibility to derive a more quantitative measure of the dust burden than FR. Thus it is necessary to carefully examine the data quality (especially the temporal consistency) of ww reports in the following aspects: (1) is the present weather observed in uniform schedule or frequency over time (addressed by Fig. 1); (2) are the dust codes being used consistently over time according to the WMO definition? (addressed by Fig. 2); and (3) do the dust codes represent the same level of dust burden consistently over time (addressed by Fig. 3 and Table 2).*

3. Pretty much all the tables and figures have NO description of what is in the table and figures in the figure captions. Please describe what is actually plotted clearly. I probably misunderstood most of the plots and would need to re-review once the plots are explained.

*Response: see my responses to your specific comments below.*

4. The author wants to indicate that drought caused some of the dust, but doesn't show statistical studies of this relationship. Please add in any mechanisms that you want to evaluate and make sure your results are statistically significant.

*Response: Section 4.2 (last 2 paragraphs) includes the analysis on the dust-drought relationship, including the correlation and significance test results between dustiness and drought index, as well as a brief discussion on the linkage of drought, vegetation, and dust emission.*

"Figure 1. Analysis of the continuity of present weather (ww) reports in ISD. (a) Global monthly number of stations (Nstn), number of ww reports (Nww), and number of dustevent reports (Ndu). For clarity Ndu is multiplied by 30; (b) Nww from different report types."

"Therefore, the SYNOP data between 1986 and 2019 are most suitable for global-scale dust analysis." I assume this means you will only present the SYNOP data in this paper. This figure can go into the supplement, and say this in the methods instead. But wait, you are using the other times periods, after you say that you shouldn't use them? Seems odd??? Please justify their use, and then discuss how that changes your results whenever you use the data that isn't very consistent.

*Response: See my response to #2 above. SYNOP data between 1986 and 2019 are considered most suitable for several reasons: (1) the number of ww reports are continuous (see Fig. 1); (2) the usage of most dust codes (except 09) is continuous (with no artefact from the alteration of code definition), especially the most frequent ww=06, 07 and the codes of dust storm events (ww=30-35) (see Fig. 2); and (3) the quantitative meaning of dust codes are nearly consistent over time (see Fig. 3 and Table 2). Generally speaking, to use any long-term measurements for variability and trend analysis of a physical phenomenon, the data quality must be carefully examined, especially in terms of temporal consistency and continuity.*

Table 2 is really unreadable because we can't understand what these codes are. I think in Table 1 you should come up with some short acronym that describes each of these codes, and use it for

the rest of the paper, so that we know which code is which and what they mean. Or just exclude this whole section, as not really very important or interesting compared to the others.

*Response: See my responses to #2 and #4 above.*

"Figure 2. Analysis of the continuity of dust weather code usage. From top to bottom: monthly number of reports of all and individual dust codes in the global SYNOP data. Horizontal lines are all-time averages. Dust weather codes are described in Table 1". This figure caption does not tell us what is plotted. Please describe the variable plotted. Is this the number of observations? Why do you present so many? Are they actually different? The same? If they are the same, then show one. If they are different, tell us how they are different and why.

*Response: See my responses to #2 and #4 above.*

"Figure 3. Temporal consistency of the harmonic mean visibility associated with dust weather codes. Dash lines are all-time averages. Dust weather codes are described in Table 1." What variable is presented in figure3? It should be described in the figure caption. Is this the number of observations? Or the correlation? Do you really need to present all the variables, or could you just show one and the offset between them?

"Figure 3 shows that while there are significant year-to-year fluctuations in the harmonic mean visibility associated with dust codes, they generally fall into three clusters: ww =06–09 (3.7 km), ww = 30–32, 98 (1.5 km), and ww = 33–35 (0.7 km)." please tell use why this is important. Does this make sense? Etc. right now there is no context for this statement.

*Response: It means that if the dust burden represented by the qualitative reports changes from year to year, and it's important to check if there are significant trends which will distort the use of dust weather reports in tracking long-term dust changes. The dust codes fall into three clusters, indicating that they contain some quantitative information of the dust burden, albeit being qualitative reports.*

"Figure 4. Decadal mean dust event frequency (FR). Gray dots are stations with FR< 1%." Please indicate in the figure caption exactly which data you use to define this, since there are many ways to do this. Repeat information in the methods to be clear. Your figure captions are so brief as to make reading your paper much more difficult than it needs to be.

*Response: FR is defined in Eq. 1. The Fig.4 caption now reads "Decadal mean dust event frequency (f, %), defined as the percentage of dust events out of the manual present weather reports. Gray dots are stations with f < 1%. The number of stations (including those with f < 1%) is shown."*

"Using Eq. 1, the decadal mean FR is calculated for stations with at least 5 years'data in each decade, as shown in Fig. 4." Please don't write sentences like this that are difficult to understand. Tell us again what equation 1 uses to calculate the frequency of occurrence in English. (for example: Using dust frequency as calcualted from the visibilty data (equation 1), we evaluate the decadal values...)

*Response: Again, FR is defined in Eq. 1. It is derived from the present weather report, not visibility. The Fig. 4 caption will be revised to clarify that (see the response above).*

These two paragraphs are very hand wavey, but on important, and easily plottable points: "The distinctive changes of decadal mean FR can be linked to multidecadal climate variations, especially the occurrence of mega-drought events lasting several years or even decades. Elevated dust activity can be observed in areas affected by persistent drought, where the reduction of soil moisture and vegetation leaves the ex-posed, dry soil prone to wind erosion. For example, a striking feature in the 1950s is the widespread, frequent dust events in the U.S. Southwest and Midwest, with several stations reporting FR > 20% in the High Plains of Texas and Colorado. The heightened dust activity was fueled by a 11-year-long (1946–1956) drought that afflicted a massive area centered in the Southwest U.S. (Fye et al., 2003). The 1950s drought was characterized by a prolonged lack of precipitation and excessive warm temperatures, which caused crop failure and livestock feed shortage (Goudie and Middleton, 1992). As the drought came to an end in the spring of 1957, FR started to decline and has since remained low in the last 50 years. Similarly, North Africa experienced progressively drier conditions during the 1970–80s in the Sahel, a semiarid dryland belt at the southern border of Sahara Desert (Giannini et al., 2008). The Sahelian drought was triggered by anomalous sea surface temperature (SST) in the tropic Atlantic and Indian Ocean (Dai, 2011). The Sahelian dust frequency during drier-than-normal years, especially in the 1980s when drought was most severe, is significantly higher com-pared to the pre- or post-drought periods. The drought-induced dust enhancement is also evident from the frequent dust weather observed downstream, including the Caribbean, Gulf of Mexico, and Iberian Peninsula. This is consistent with the long-term in situ dust measurements in Barbados and Miami, Florida, indicating a positive cor-relation between the Sahel dry anomaly and African dust outflow across the tropical North Atlantic (Prospero and Lamb, 2003; Zuidema et al., 2019). With the amelioration of Sahelian drought in the 2000s, FR experienced significant decreases at the source and downwind, consistent with ground and satellite observations (Hsu et al., 2012; Li etal., 2014). In the past decade, increased dust activity can be observed in West Africa and the Middle East, which will be discussed later." Please show this is true statistically significantly, in a clear way. perhaps show the 1-sigma and 2-sigma bounds and highlight the time periods above or below, or do a correlatoin.

*Response: The discussions presented here are simply based on the visual examination of the decadal mean FR and VI from the 1950s to the 2010s. The discussions are connected to and supported by a number of previous studies cited in the paper. Please be specific about the 'hand waving'.*

"if weather stations provide a consistent view of global dust variations, FR and VI are compared with the datasets described in Sect. 2.3." You can compare the dust variations to satellite data and AERONET data and evaluate the dust variables, but please do not pretend that MERRA output is more than model output.

*Response: No pretension is made here. Section 2 clearly describes CAMS and MERRA2 as global model reanalysis. Although they are model output with uncertainties, the correlation*

*analysis in Table 3 shows good agreement between surface stations, MODIS, and model reanalysis in terms of the long-term variability.*

"Figure 6. Global monthly (gray) and annual (black dotted) FR and VI . Horizontal lines are all-time averages." Please describe what you are plotting in complete detail, including repeating which variables are used, etc, so that your figures are self-standing. How are you averaging over the globe when you only have spotty data? Needs to be described in the methods section, and make sure you are doing this in a manner that is consistent with the observations and models you are comparing again (for example, pick each point from the dust station data, and match to the model output at the same grid box, so are weighting similarly).
*Response: The global mean FR and VI are derived by aggregating all global stations together. The station map is given in Fig. S1, S2, and is also shown in Fig. 4 and 5. The stations have fairly good coverage of the major dust source areas. Similarly, the global mean DOD is calculated from MODIS (Voss and Amato, 2020), CAMS, and MERRA2 without taking into account coincidence in space and time, as one normally would do in using these data in analyzing the global mean. Collocation would be needed for a critical comparison of MODIS vs. reanalysis, which is outside the scope of this study.*

"Trends are further calculated from the monthly anomalies using the pyMannKendallpackage developed by Hussain and Mahmud (2019), which consists of multiple Mann-Kendall test options to accommodate the seasonality and serial correlation in the data. The Mann-Kendall test is a non-parametric test of the presence of monotonic trend in the data, and has advantage over parametric methods (e.g., t test) for its in-sensitivity to outliers, missing values, and the statistical distribution of the data. TheMann-Kendall test is designed for serially independent data and thus can be influenced by the presence of autocorrelation in the data, which either increases the uncertainty of estimated trends or prolongs the length of time period required to detect a given trend (Weatherhead et al., 1998)." All methods should be in the methods section, or in figure captions, not in the results section.

*Response: My preference is to briefly describe the trend detection method here, rather than using a separate subsection.*

"The decadal mean FR and VI (Fig. 4 and 5) indicate mega-drought events are associated with extremely active dust periods in the 20th century." To make this statement you need to have compared against precipitation or P-E data and show a statistically significant change/relationship

*Response: This statement repeats the finding from the discussions in Section 4.1, which is also supported by a number of previous studies cited in the manuscript, e.g., Goudie and Middleton, 1992; Prospero and Lamb, 2003.*

"Dust variation in North Africa from 1986 to 2019. From top to bottom: monthly Nwwand Ndu, and anomalies of FR, VI, MERRA2 DOD, wind speed, soil moisture, and theJones North Atlantic Oscillation (NAO) index. DOD and soil moisture are averaged overregion 10◦N–20◦N, 20◦W–20◦E; Dotted curves are annual averages. Dashed lines areTheil-Sen linear regression of

FR, VI and DOD, with the annual trends and confidence intervals (CI) shown." What is Nww? Ndu? Should be explained in the figure caption, as well as FR, VI.

*Response: The figure caption now reads "Dust variation in North Africa from 1986 to 2019. From top to bottom: number of present weather reports (Nww), number of dust event reports (Ndu, multiplied by 3 for clarity), and the anomalies of dust event frequency (f), dust extinction coefficient (β), MERRA2 dust optical depth (DOD), wind speed, soil moisture, and the Jones North Atlantic Oscillation (NAO) index. DOD and soil moisture are averaged over region 10°N–20°N, 20°W–20°E. Gray curves are monthly data, while dotted black curves are annual averages. Dashed lines are Theil-Sen linear regression of f, β and DOD, with the annual trends and confidence intervals (CI) shown.".*

"Consistent with previous studies, Fig. 11 shows that African dust was at a historic high during the late 1980s due to abnormally strong winds and low soil moisture, both favorable for dust production (e.g., Prospero and Lamb, 2003; Chiapello et al., 2005; Evan et al., 2016)." Is this statistically significantly true? Please check or put your statistical significance on the plot.

*Response: As shown in Fig. 11, the monthly FR and VI anomalies are well above the zero mean and at a maximum in the 1980s and early 1990s, in agreement with the past studies cited in the manuscript.*

"Indeed, significant correlations are found between NINO 3.4 and scPDSI (r = 0.27,p < 0.001), as well as between PDO and scPDSI (r = 0.53, p < 0.001). PDO also appears to be more correlated (r =−0.36, p < 0.001) with MERRA2 DOD than NINO3.4 (r =−0.15, p < 0.005). As seen in Fig. 13, PDO has transitioned into positive and weak negative phases since 2015, which has led to the amelioration of drought and consequently, a decrease of dust activity in the Middle East." Is this based on the time series in the plots? Please specify.

*Response: It is based on the data shown in Fig. 13. NINO 3.4 covaries with PDO and is not shown in Fig. 13, though.*

"The dust decline was driven by the weakening of surface winds, as shown in Fig. 15 and suggested by past studies, which reported widespread decrease of surface winds in China during the 1970s– 1990s, with greatest reduction during spring and in the gusty wind segment (Guo et al., 2011; Lin et al., 2013)." Again, do not show your results without showing that they are statistically significant. Please show a correlation coefficient and that it is significant before you make such important, but unclear statements.

*Response: These statements are backed by both previous studies and statistical analysis (with significance test results) presented in section 5.3 (see line 448-476).*